# Galileo: Learning Global & Local Features of Many Remote Sensing Modalities

**Gabriel Tseng** [* 1 2 3]  **Anthony Fuller** [* 4]  **Marlena Reil** [1 2]  **Henry Herzog** [3]  **Patrick Beukema** [3]  **Favyen Bastani** [3]
**James R. Green** [4]  **Evan Shelhamer** [4 5 6]  **Hannah Kerner** [† 7]  **David Rolnick** [† 1 2]

## Abstract

We introduce a highly multimodal transformer to represent many remote sensing modalities—multispectral optical, synthetic aperture radar, elevation, weather, pseudo-labels, and more—across space and time. These inputs are useful for diverse remote sensing tasks, such as crop mapping and flood detection. However, learning shared representations of remote sensing data is challenging, given the diversity of relevant data modalities, and because objects of interest vary massively in scale, from small boats ($1-2$ pixels and fast) to glaciers (thousands of pixels and slow). We present a novel self-supervised learning algorithm that extracts multi-scale features across a flexible set of input modalities through masked modeling. Our dual global and local contrastive losses differ in their targets (deep representations vs. shallow input projections) and masking strategies (structured vs. not). Our **Galileo** is a single generalist model that outperforms SoTA specialist models for satellite images and pixel time series across eleven benchmarks and multiple tasks.

## 1. Introduction

Learning representations of large-scale and multimodal remote sensing and geospatial data is a long-standing scientific and practical goal. This goal is motivated by the increasing impact of machine learning (ML) and remote sensing (RS) in societally important domains (e.g. food security (Kerner et al., 2020) and disaster response (Frame et al., 2024)) where labels are expensive or difficult to acquire (Kebede et al., 2024).

Self-supervised learning (SSL) can make it possible to har-

*Equal contribution †Equal Supervision ¹Mila – Quebec AI Institute ²McGill University ³Allen Institute for AI (Ai2) ⁴Carleton University ⁵University of British Columbia ⁶Vector Institute ⁷Arizona State University. Correspondence to: Gabriel Tseng <gabriel.tseng@mail.mcgill.ca>.

*Proceedings of the 42ⁿᵈ International Conference on Machine Learning*, Vancouver, Canada. PMLR 267, 2025. Copyright 2025 by the author(s).

ness vast quantities of unlabeled data, as is available in the case of remote sensing. However, SSL methods for RS (Jean et al., 2019; Manas et al., 2021) have been specialized to certain input modalities or shapes, such as pixel time series vs. image time series, following pioneering methods for learning from images (Chen et al., 2020; He et al., 2022) and text (Devlin et al., 2018). In a nutshell, these methods create two versions ("views") of an input and *pre*train models to predict one view given the other. After pretraining, the learned representations can then transfer to real tasks through finetuning or reuse as features, even with limited labels or computation. We unify SSL across multiple modalities and input shapes used for remote sensing in practice, yielding a *flexible* model of both image and pixel time series.

For spatiotemporal scale, satellite imagery encompasses objects of a variety of spatial and temporal extents. Common resolutions are 10m per pixel and 6 acquisitions per month. Thus — unlike in most natural imagery (e.g., ImageNet (Deng et al., 2009)) or video (e.g., Kinetics-400 (Kay et al., 2017)) — an object in RS (such as a small fishing boat) may be represented by only a *single* pixel in RS and can be present in just a *single* frame (Beukema et al., 2023). Conversely, an object may be a kilometer-scale glacier that requires tracking over decades (Baraka et al., 2020).

A second challenge is presented by the number and variety of sensors used in RS, where most methods have only limited flexibility in the data types on which they operate. Many methods model multispectral optical (MS) data (Cong et al., 2022; Noman et al., 2024; Nedungadi et al., 2024), synthetic aperture radar (SAR) data (Wang et al., 2024b;a), or joint MS and SAR data (Fuller et al., 2024; Xiong et al., 2024), but not other modalities and not across time. Other methods model MS data across time, but no other modalities (Bastani et al., 2023; Szwarcman et al., 2024). Limiting the number and diversity of views of the Earth for learning may limit the utility and generality of the resulting representations for predictions and analysis. This could limit transfer with or without finetuning, and especially without, which may be more computationally feasible for applied and interdisciplinary practitioners.

To address both of these challenges, we propose **Galileo**, a new family of models for multiple modalities (optical,

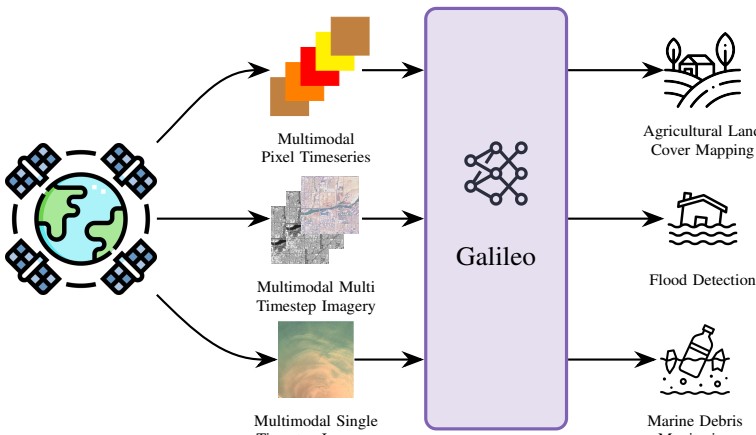

*Figure 1.* A *single* Galileo model can be applied to a wide range of remote sensing tasks. We achieve this by training Galileo on the diversity of remote sensing modalities used by practitioners for different applications. In addition, we train Galileo to process *views* of these modalities used by practitioners, ranging from pixel time series to multi timestep imagery to single timestep imagery.

radar, etc.), scales, and shapes (pixel time series, image time series, single images) of remote sensing data. Our models learn *multimodal, multi-scale, and flexible* representations for Earth observation with SoTA downstream task results. We achieve this with (i) a novel self-supervised learning (SSL) algorithm which extends the masked data modeling framework to learn useful representations of "local" and "global" features, and (ii) a globally sampled, highly multimodal pretraining dataset which includes inputs specifically selected because of their use across diverse remote sensing tasks.

We demonstrate Galileo's accuracy on an extensive suite of benchmarks, covering many applications, domains, and RS data types. Specifically, our Galileo-Base model ranks first above larger RS models specialized for images, such as SatMAE (Cong et al., 2022) and CROMA (Fuller et al., 2024), and with the same set of weights Galileo-Base ranks above RS models specialized for pixel time series such as Presto (Tseng et al., 2023).

## 2. Global, Local, Multimodal Self-Supervision

We collect a large, rich dataset of highly multimodal remote sensing data specifically sampled for geographic and semantic diversity (Sec. 2.3). To learn rich representations of the diverse modalities in this dataset across different spatiotemporal scales, we design a novel and highly effective SSL algorithm to simultaneously learn local and global features.

### 2.1. Method Intuition

Galileo learns representations via *two* masked data modeling objectives: our **global** and **local** tasks (Figure 2). Masked modeling takes an input $\mathbf{x}$, divides it into masked "targets" $\mathbf{x}_t$ and a "visible" view $\mathbf{x}_v$, then predicts the masked part from the visible part. We leverage a transformer architecture for masked modeling of deep and shallow representations of remote sensing data. As a transformer, this model requires

tokenization of its inputs (Sec. 2.2.1). Our masking and prediction are performed over latent tokens and not on the pixels.

Our **global** and **local** objectives differ in both targets and masking, as we now detail.

**Deep targets → global features; shallow targets → local features.** Our target prediction occurs in the latent space, so we construct target tokens by passing our target input $\mathbf{x}_t$ to a "frozen" encoder (Sec. 2.2.3). We construct **global** targets by processing our target input $\mathbf{x}_t$ with our frozen encoder. We construct **local** targets by processing our target input $x_t$ with a minimal linear layer to match dimensionality. **Intuitively, deeper representations are more global due to more non-local processing by attention, while shallower representations are more local due to less processing of the inputs.** Galileo learns representations of both global and local features by simultaneously pretraining on deep and shallow targets.

**Space-time masking → global features; unstructured masking → local features.** Masking determines which tokens are visible, i.e., which are used as inputs and which are used as target outputs (Sec. 2.2.2). The choice of masking matters for the learned representations by governing the type and difficulty of the predictions needed. **Intuitively, larger masks require larger or more global predictions, while smaller masks require smaller or more local predictions.**. We thus specialize **global masking** to divide visible and target tokens by larger spans, with correlated "space-time" masking, and specialize **local masking** to divide visible and target tokens uniformly at random. Galileo learns multi-scale features by simultaneously applying global structured masking (longer spans) and unstructured local masking (shorter spans) during pretraining.

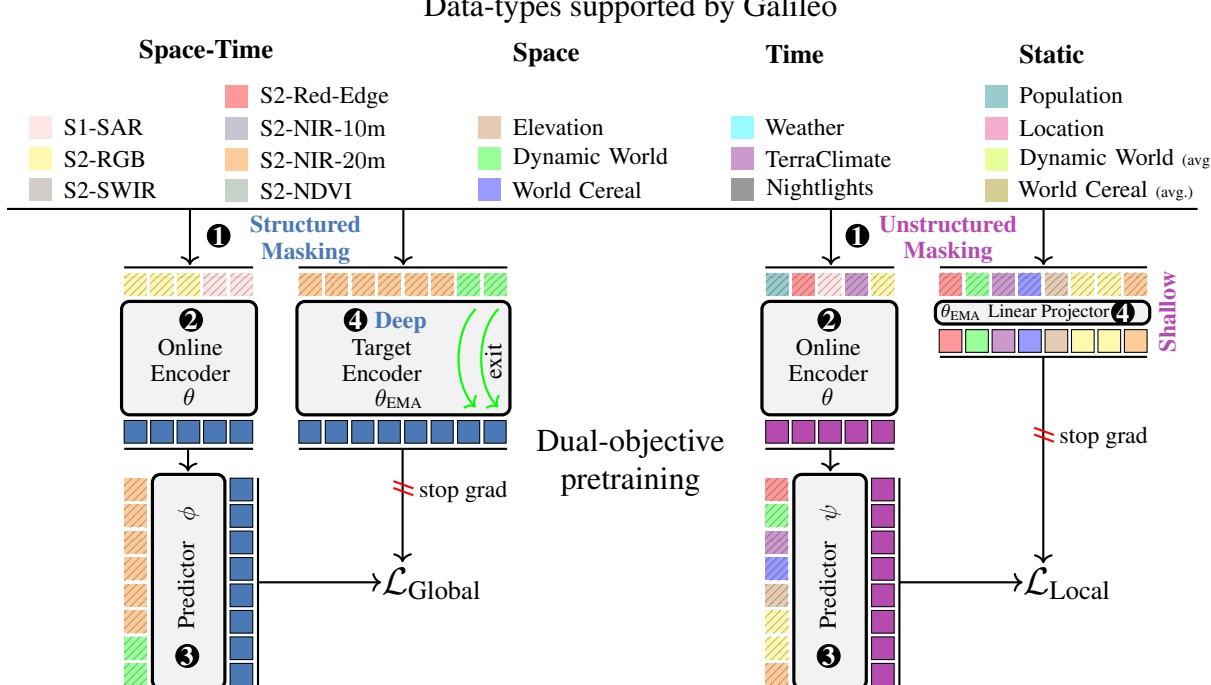

*Figure 2.* We train Galileo with our **global (left)** and **local (right)** pretraining losses. Black-outlined tokens are model outputs, black-striped tokens are model inputs. Steps: ❶ sample from dataset and mask (**structured left**, **unstructured right**), ❷ encode "visible" tokens, ❸ predict targets given target queries and visible encodings, ❹ encode targets (**deep left**, **shallow right**) with stop gradient, and ❺ calculate within-sample token contrastive loss.

## 2.2. Method Details

### 2.2.1. INPUT TOKENIZATION

Our vision transformer (ViT)-based architecture requires tokenization to convert remote sensing inputs into a series of tokens. Our encoder splits the input into spatial squares, timesteps, and channel groups (related subsets of channels from a remote sensing product (e.g. one channel group for the 10m channels in Sentinel-2 data)). It then projects these inputs to the encoder dimension $D$ using the following transformations: (i) Space-time data, $\mathbb{R}^{H \times W \times T \times C} \to \mathbb{R}^{\frac{H}{P} \cdot \frac{W}{P} \cdot T \cdot G \times D}$, $H$ is the height, $W$ is the width, $P$ is the patch size (in pixels per side), $T$ is the timesteps, $C$ are the channels, $G$ are the channel groups. (ii) Space data, $\mathbb{R}^{H \times W \times C} \to \mathbb{R}^{\frac{H}{P} \cdot \frac{W}{P} \cdot G \times D}$, (iii) Time data, $\mathbb{R}^{T \times C} \to \mathbb{R}^{T \cdot G \times D}$, and (iv) Static data, $\mathbb{R}^{C} \to \mathbb{R}^{G \times D}$.

**Token Embeddings.** After these linear projections, our encoder creates spatial and temporal sinusoidal position embeddings, learnable channel embeddings, and month embeddings to enable seasonal reasoning; we denote these token position embeddings as $\mathbf{e} \in \mathbb{R}^{L \times D}$, where $L$ is the token sequence length. Our encoder adds these embeddings to the already-computed linear projections. It concatenates all channel groups along the sequence dimension — forming our input sequence, $\mathbf{x} \in \mathbb{R}^{L \times D}$.

Apart from our custom tokenization and use of resizable linear projection weights (Beyer et al., 2023) our transformer architecture remains compatible with standard ViTs. This makes it simple to use and highly flexible w.r.t. input sequence length, the various channel groups, and other differences across our various sources of data.

### 2.2.2. CONSTRUCTING INPUTS VIA MASKING

Given an input $\mathbf{x}$, we construct a "visible" view $\mathbf{x}_v \in \mathbb{R}^{L_v \times D}$ and a "target" view $\mathbf{x}_t \in \mathbb{R}^{L_t \times D}$. For both global and local tasks, the goal is to predict the target tokens given the visible tokens. However, our masking strategies (i.e., rules that govern view construction) differ between tasks.

**Global features via space and time masking.** "Space masking" randomly samples tokens across space while maintaining consistency across time; "time masking" randomly samples tokens across time while maintaining consistency across space. In both cases, we select modalities to be encoded *or* decoded. This strategy increases the distance between visible and target tokens.

**Local features via unstructured masking.** Unstructured masking randomly samples tokens with the same probability regardless of their space, time, or channel group position. This strategy minimizes the average distance between visi-

*Table 1.* When compared to existing pretrained remote sensing models, the Galileo models are both the best performing and most flexible models. **Performance** is measured via rankings (where lower numbers are better) on image tasks in Tables 15, 16 & 17 and pixel-timeseries tasks in Table 6. For clarity, we select the best architecture per method; full rankings are available in Table 18. **Flexibility** is measured by documenting which inputs are supported by the models: MultiSpectral (MS), Synthetic Aperture Radar (SAR), additional Remote Sensing modalities (+modalities), inputs with spatial dimensions and inputs with more than 1 or 4 timesteps. Galileo-Base is the best performing model compared to both image-specialized models (e.g. CROMA) and pixel-timeseries specialized models (e.g. Presto).

| Method | Arch. | Rank ↓ Images | Rank ↓ Pixel-timeseries | MS | SAR | +modalities | Spatial dims | > 1 timestep | > 4 timesteps |
|---|---|---|---|---|---|---|---|---|---|
| SatMAE | ViT-Large | 10.4 | N/A | ✔ | | | ✔ | | |
| SatMAE++ | ViT-Large | 10.9 | N/A | ✔ | | | ✔ | | |
| CROMA | ViT-Base | 4.3 | N/A | ✔ | ✔ | | ✔ | | |
| SoftCon | ViT-Base | 5.9 | N/A | ✔ | ✔ | | ✔ | | |
| DOFA-v1 | ViT-Large | 9.4 | N/A | ✔ | ✔ | | ✔ | | |
| Satlas | Swin-Tiny | 12.9 | N/A | ✔ | | | ✔ | ✔ | |
| MMEarth | CNN-atto | 12.3 | N/A | ✔ | | | ✔ | | |
| DeCUR | ViT-Small | 8.3 | N/A | ✔ | ✔ | | ✔ | | |
| Prithvi 2.0 | ViT-Large | 11.7 | N/A | ✔ | | | ✔ | ✔ | |
| AnySat | ViT-Base | 11.1 | 4.5 | ✔ | ✔ | ✔ | ✔ | ✔ | ✔ |
| Presto | ViT-Presto | N/A | 3.0 | ✔ | ✔ | ✔ | | ✔ | ✔ |
| **Galileo** | ViT-Nano | 10.9 | 3.5 | ✔ | ✔ | ✔ | ✔ | ✔ | ✔ |
| **Galileo** | ViT-Tiny | 6.4 | 2.3 | ✔ | ✔ | ✔ | ✔ | ✔ | ✔ |
| **Galileo** | ViT-Base | **3.0** | **1.8** | ✔ | ✔ | ✔ | ✔ | ✔ | ✔ |

ble and target tokens.

### 2.2.3. ENCODING VISIBLE AND TARGET TOKENS

**Inputs.** Our "online" encoder computes encodings for the visible tokens, $\mathbf{z}_v = \mathbf{E}(\mathbf{x}_v)$. This model's parameters are updated via gradient descent.

**Targets.** Our "target" encoder computes encodings for the target tokens, $\mathbf{z}_t = \mathbf{E}_{\text{EMA}}(\mathbf{x})$. This model's parameters are updated via computing the exponential moving average of the online encoder; this use of EMA is common in SSL (Chen et al., 2021; Assran et al., 2023). Unlike prior work, Galileo training chooses different depths (encoder layers) for the targets depending on the task (**global** vs. **local**).

**Global features via deep targets.** We target the token representations after the $\ell^{\text{th}}$ layer, where $\ell$ varies by modality. We select $\ell$ based on the level of processing for each modality: pseudo-labels use only linear projections (no encoder layers), Sentinel-1 and Sentinel-2 use *all* encoder layers, and other channels use half the encoder layers. We denote our level-specific target encoder as $\mathbf{E}_{\text{EMA}}^{\ell}$.

**Local features via shallow targets.** We target the lowest representation level: the input space. For compatible dimensionality, we compute targets using the target encoder's linear projection, $\mathbf{E}_{\text{EMA}}^{proj}$ which maps all tokens to the embedding dimension $D$. This strategy *skips all deeper processing.*

### 2.2.4. MAKING PREDICTIONS & COMPUTING LOSSES

A predictor transformer $\mathbf{P}$ receives the position, time, month, and channel group embeddings $\mathbf{e}_t$ for the target tokens and predicts patch encodings $\mathbf{p}_t$ by cross-attending to the visible encodings, i.e., $\mathbf{p}_t = \mathbf{P}(\mathbf{e}_t, \mathbf{z}_v)$. Finally, the predictions $\mathbf{p}_t$ and targets $\mathbf{z}_t$ are compared to compute a loss $\mathcal{L}(\mathbf{p}_t, \mathbf{z}_t)$ that updates the online encoder.

We use the "Patch Discrimination" loss (PatchDisc (Wei et al., 2024)) for both tasks, which applies the InfoNCE loss between tokens *within* an input:

$$\mathcal{L}(\mathbf{u}, \mathbf{v}) = \frac{1}{L_i} \sum_j^{L_i} \log \frac{\exp(\text{sim}(\mathbf{u}_{i,j}, \mathbf{v}_{i,j})/\tau)}{\sum_j^{L_i} \exp(\text{sim}(\mathbf{u}_{i,j}, \mathbf{v}_{i,j})/\tau)}$$

with the softmax temperature $\tau$, the input index $i$, the token index $j$, the number of tokens in the $i^{th}$ input $L_i$, and the $l_2$ normalized dot product $\text{sim}(\mathbf{u}, \mathbf{v}) = \mathbf{u}^{\top}\mathbf{v}/\|\mathbf{u}\|\|\mathbf{v}\|$.

**Amplifying local features via input-contrastive learning.** We design a challenging self-supervised task by applying PatchDisc to shallow representations of inputs by linear projections of the pixels. The predictor must output tokens that are similar to the pixels at the same positions but dissimilar to pixels at other positions. This differs from reconstruction methods, like MAE (He et al., 2022), which predict pixels (via the mean-squared error), but do not repel other pixels in the sequence. This differs from joint embedding methods, like LatentMIM (Wei et al., 2024) or I-JEPA (Assran et al., 2023), which target deep representations only.

Finally, we average the **global** and **local** losses:

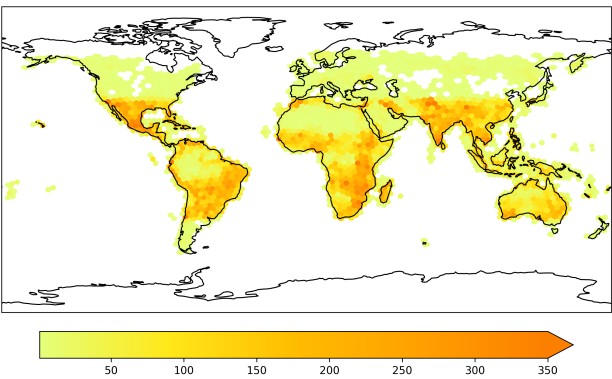

*Figure 3.* The number of training points per H3 cell (Uber, 2018) at resolution = 2. We sample from the entire globe for geographic and semantic diversity (as measured by WorldCover classes (Zanaga et al., 2022)).

*Table 2.* Galileo's combined algorithm retains the within-input diversity of our local algorithm and achieves a between-input diversity between our global and local algorithms. Diversity is measured as $1 - $ cosine similarity over the EuroSat training data.

| Pretraining Objective | Within-input Diversity | Between-input Diversity |
|---|---|---|
| Global only | 0.10 | 0.25 |
| Local only | 0.34 | 0.14 |
| Combined | 0.35 | 0.19 |

$$\mathcal{L}_{Galileo} = \tfrac{1}{2}(\mathcal{L}_{Global} + \mathcal{L}_{Local})$$

### 2.2.5. MEASURING LEARNED REPRESENTATIONS

We experiment to verify our intuition about global and local tasks: the **global** task should encourage **between**-input feature diversity and our **local** task should encourage **within**-input feature diversity. For all EuroSat training inputs, we measure how features differ locally *within* an input by computing cosine similarities of token representations $\mathbf{z} \in \mathbb{R}^{L \times D}$. Similarly, we measure how features differ globally *between* inputs by computing cosine similarities of global-averaged token representations $\mathbf{z} \in \mathbb{R}^{D}$. Our local task amplifies within-input features, while our global task amplifies between-input features (Table 2). We confirm these intuitions on downstream tasks in Section 4.1.

### 2.3. Galileo's Pretraining Data

We collect a large, global pretraining dataset of 127,155 instances. Figure 3 maps the training points. We include a wide range of RS inputs to serve diverse applications. Each instance consists of 4 types of data covering 9 RS data modalities. We select the modalities by their uses in machine learning for remote sensing efforts (Van Tricht et al., 2023; Beukema et al., 2023; Poggio et al., 2021). Section B.1 describes our full dataset sampling process.

We group the modalities by whether they vary in space, time,

both, or neither. A single instance consists of 24 monthly timesteps and $96 \times 96$ pixels at a 10m/pixel resolution.

**Space-time varying data.** These data consist of imagery acquired by Sentinel-1 & -2 satellites. For Sentinel-1, we take the VV and VH polarizations; and for Sentinel-2, we take all bands except the B1, B9 and B10 bands. All bands are resampled to a 10m/pixel resolution. We also include NDVI (Tucker, 1979) from Sentinel-2 as an input.

**Space varying data.** These data consist of elevation and slope captured by the Shuttle Radar Topography Mission (NASA JPL, 2000), which are constant in time; Dynamic World land cover map probabilities (Brown et al., 2022), averaged over time for temporal consistency; and World Cereal agricultural land cover maps (Van Tricht et al., 2023).

**Time varying data.** These data consist of precipitation and temperature from the ERA5 dataset (Hersbach et al., 2020); climate water deficit, soil moisture, and actual evapotranspiration from TerraClimate (Abatzoglou et al., 2018); and VIIRS nighttime lights (Elvidge et al., 2017). Although these modalities vary in space as well, their spatial resolution (ERA5 has a spatial resolution of tens of kilometres per pixel) means we treat them as static in space from the perspective of a single instance.

**Static data.** These data consist of population estimates from the LandScan dataset (Dobson et al., 2000), the spatial location of the instance, defined by its central latitude and longitude, Dynamic World classes spatially averaged over the instance, and World Cereal agricultural land cover maps spatially averaged over the instance. We include the averaged Dynamic World and World Cereal inputs in addition to the space-varying inputs.

## 3. Experimental Framework

**Pretraining.** We pretrain three model sizes for 500 epochs using the algorithm described in Section 2.2. Please see the Appendix B.2 for complete details.

**Downstream Tasks.** We evaluate our model on all Sentinel-2 tasks in GeoBench (Lacoste et al., 2024). These cover single-timestep image classification and segmentation in various applications and geographies. We also test on fine-grained segmentation via the MADOS marine debris dataset (Kikaki et al., 2024), Sentinel-1 image segmentation via Sen1Floods11 (Bonafilia et al., 2020), image time series segmentation via PASTIS (Garnot & Landrieu, 2021), optical pixel time series classification via Breizhcrops (Rußwurm et al., 2019), and multimodal pixel time series classification via CropHarvest (Tseng et al., 2021).

**Comparisons.** We benchmark Galileo against all SoTA pretrained RS models (described in Section 5). We report results on the full test set for each task. Input normaliza-

| Method | Arch. | m-EuroSat Top-1 Acc. Training % | | m-BigEarthNet F1 Score Training % | | m-So2Sat Top-1 Acc. Training % | | m-Brick-Kiln Top-1 Acc. Training % | |
|---|---|---|---|---|---|---|---|---|---|
| | | 100% | 1% | 100% | 1% | 100% | 1% | 100% | 1% |
| SatMAE | ViT-Base | 84.1 | 34.8 | 50.6 | 29.0 | 36.0 | 23.1 | 86.1 | 73.5 |
| SatMAE++ | ViT-Large | 82.7 | 48.5 | 50.8 | 31.6 | 34.7 | 23.4 | 89.6 | 76.7 |
| CROMA | ViT-Base | 85.6 | _51.3_ | 58.8 | _44.7_ | 48.8 | 33.8 | **92.6** | **85.1** |
| SoftCon | ViT-Small | 89.8 | 27.2 | **64.7** | 43.3 | _51.1_ | 31.4 | 89.2 | 77.8 |
| DOFA-v1 | ViT-Base | 82.8 | 49.6 | 49.4 | 29.9 | 41.4 | 29.4 | 88.3 | 78.3 |
| Satlas | Swin-Tiny | 81.7 | 35.8 | 51.9 | 29.6 | 36.6 | 27.1 | 88.2 | 73.0 |
| MMEarth | CNN-atto | 81.7 | 30.0 | 58.3 | 39.6 | 39.8 | 25.1 | 89.4 | 79.7 |
| DeCUR | ViT-Small | 89.0 | 46.6 | _63.8_ | **49.6** | 45.8 | 30.9 | 83.7 | 74.2 |
| Prithvi 2.0 | ViT-Large | 80.2 | 48.0 | 49.4 | 28.8 | 29.5 | 26.1 | 87.9 | _80.6_ |
| AnySat | ViT-Base | 82.2 | 47.1 | 54.9 | 33.7 | 39.8 | 29.0 | 85.3 | 72.0 |
| Galileo | ViT-Nano | 89.7 | 41.7 | 53.8 | 33.9 | 50.1 | _37.4_ | 86.7 | 79.7 |
| Galileo | ViT-Tiny | 90.1 | 41.3 | 55.5 | 34.4 | 49.7 | 36.2 | 86.9 | 77.3 |
| Galileo | ViT-Base | **93.0** | **56.6** | 59.0 | 36.5 | **54.8** | **43.2** | 90.7 | 78.0 |

*Table 3.* Galileo-Base is the best model for image classification (%) by $k$NN. We show the best architecture per method. We **bold** and underline the 1st and 2nd best results across all methods and architectures, as reported in Table 15.

| Method | Arch. | m-EuroSat Top-1 Acc. Training % | | m-BigEarthNet F1 Score Training % | | m-So2Sat Top-1 Acc. Training % | | m-Brick-Kiln Top-1 Acc. Training % | |
|---|---|---|---|---|---|---|---|---|---|
| | | 100% | 1% | 100% | 1% | 100% | 1% | 100% | 1% |
| SatMAE | ViT-Large | 96.6 | 56.9 | 68.3 | 41.8 | 57.2 | 36.4 | 98.4 | 96.1 |
| SatMAE++ | ViT-Large | 96.5 | 56.4 | 67.9 | _45.6_ | 56.0 | 36.9 | 98.6 | 92.5 |
| CROMA | ViT-Large | 96.6 | 52.7 | _71.9_ | **47.9** | 60.6 | 40.9 | 98.7 | 96.7 |
| SoftCon | ViT-Base | 97.5 | 56.3 | 70.3 | 38.5 | 61.7 | _49.2_ | 98.7 | **97.3** |
| DOFA-v1 | ViT-Large | 96.9 | 53.4 | 68.0 | 43.5 | 58.7 | 37.0 | 98.6 | 94.5 |
| Satlas | Swin-Base | 97.5 | 51.9 | **72.8** | 25.8 | _61.9_ | 30.6 | 98.4 | 94.7 |
| MMEarth | CNN-atto | 95.7 | 47.5 | 70.0 | 43.4 | 57.2 | 30.0 | **98.9** | 89.2 |
| DeCUR | ViT-Small | **97.9** | 54.2 | 70.9 | 44.7 | 61.7 | 47.0 | 98.7 | 96.9 |
| Prithvi 2.0 | ViT-Large | 96.5 | 51.5 | 69.0 | 37.1 | 54.6 | 31.0 | 98.6 | 96.2 |
| AnySat | ViT-Base | 95.9 | 51.3 | 70.3 | 13.3 | 51.8 | 29.7 | 98.6 | 85.6 |
| Galileo | ViT-Nano | 94.5 | 52.6 | 67.1 | 23.3 | 57.4 | 34.9 | 96.1 | 94.2 |
| Galileo | ViT-Tiny | 96.9 | _60.6_ | 69.7 | 39.5 | _61.9_ | 43.1 | 98.7 | 96.6 |
| Galileo | ViT-Base | _97.7_ | **63.5** | 70.7 | 40.9 | **63.3** | **50.6** | 98.7 | 96.8 |

*Table 4.* Galileo-Base is the best model for image classification (%) by finetuning. We show the best architecture per method. We **bold** and underline the 1st and 2nd best results across all methods and architectures, as reported in Table 16.

tion, image sizes, and hyperparameter selections impact performance (Corley et al., 2024), so we therefore re-run evaluations for all comparisons and sweep normalization methods and learning rates (where appropriate). In addition, we resize all images for each model to its pretraining image size. For the image classification and segmentation tasks, we measure results across four training set sizes ("partitions"): 100%, 20%, 5%, and 1%. We use a patch size of 4 for all models with variable patch sizes. When applying single-timestep models to the multi-timestep PASTIS dataset, we additionally sweep pooling methods to pool per-timestep representations. See Appendix C for complete details.

## 4. Results

We present model rankings averaged across all tasks and partitions in Table 1. We evaluate Galileo on common RS benchmarks. While these common RS benchmarks typically consist of a limited set of RS modalities (optical and radar data), Galileo learns from many additional modalities during pretraining. These modalities are readily available to practitioners (Table 1, "Supported Inputs"), and may deliver improvements at test time.

**Image results.** We compare Galileo to image-specialized models in Tables 3, 4 and 5. These models were pretrained on single-timestep imagery, devoting all their capacity to images, except for Satlas. Nonetheless, Galileo-Base outranks all of them on image classification and segmentation. Our small ViT-{Nano, Tiny} models also excel at these tasks, and often outperform much larger models, which is valuable for limited computation applications. Furthermore, Galileo's variable patch size can exchange computational cost and task performance: we plot this trade-off in Figure 4. By increasing the patch size, an input is split up into fewer tokens, reducing the MACs required for feature extraction.

Besides Galileo, AnySat is the only model to support both single-timestep images and pixel time series. Of these two generalist models, Galileo is the more accurate on standard benchmarks (by 10.8% on EuroSat for example). (Note: for semantic segmentation, the AnySat features are per-pixel instead of per-patch. For comparable training cost, we sample 6.25% of its pixel features per image when training, but evaluate with all pixel features when testing. We confirmed the fairness of this evaluation with the the AnySat authors by personal communication.)

**Time series results.** We compare Galileo to the generalist

| Method | Arch. | m-Cashew-Plant Training % 100% | m-Cashew-Plant Training % 1% | m-SA-Crop-Type Training % 100% | m-SA-Crop-Type Training % 1% | MADOS Training % 100% | MADOS Training % 1% | Sen1Floods11 Training % 100% | Sen1Floods11 Training % 1% | PASTIS Training % 100% | PASTIS Training % 1% |
|---|---|---|---|---|---|---|---|---|---|---|---|
| SatMAE | ViT-Large | 30.8 | 22.7 | 24.8 | 16.9 | 55.6 | 13.2 | N/A | | 29.6 | 11.5 |
| SatMAE++ | ViT-Large | 29.6 | 23.3 | 25.7 | 16.8 | 49.9 | 12.7 | N/A | | 30.5 | 12.0 |
| CROMA | ViT-Base | 31.8 | 26.8 | **32.0** | 18.3 | 64.2 | **24.4** | 78.9 | 77.6 | 44.4 | 18.5 |
| SoftCon | ViT-Base | 29.6 | 22.8 | 30.8 | 18.5 | 60.3 | 16.5 | 78.0 | 74.8 | 31.3 | 10.5 |
| DOFA-v1 | ViT-Large | 27.7 | 23.3 | 25.4 | 16.8 | 51.6 | 19.1 | 78.1 | 77.4 | 29.8 | 13.4 |
| Satlas | Swin-Tiny | 25.1 | 18.6 | 23.4 | 16.2 | 45.9 | 12.4 | N/A | | 28.0 | 10.9 |
| MMEarth | CNN-atto | 24.2 | 20.3 | 22.2 | 14.1 | 34.2 | 16.1 | N/A | | 24.0 | 10.5 |
| DeCUR | ViT-Small | 26.2 | 22.8 | 21.5 | 15.3 | 54.8 | 16.6 | 74.5 | 72.2 | 22.4 | 11.0 |
| Prithvi 2.0 | ViT-Large | 26.7 | 23.2 | 22.9 | 15.7 | 50.0 | 18.9 | N/A | | 29.3 | 13.2 |
| AnySat | ViT-Base | 26.1 | 21.7 | 27.1 | 15.8 | 50.2 | 17.0 | 77.9 | 76.9 | **46.2** | **23.5** |
| **Galileo** | ViT-Nano | 24.4 | 24.5 | 19.7 | 14.5 | 54.8 | 13.9 | 78.6 | 77.1 | 17.5 | 13.1 |
| **Galileo** | ViT-Tiny | 27.4 | 27.9 | 22.5 | 17.1 | 60.8 | 17.5 | 78.0 | 77.9 | 28.1 | 16.9 |
| **Galileo** | ViT-Base | 33.0 | **30.2** | 30.1 | **19.4** | **67.6** | 14.7 | **79.4** | **78.2** | 39.2 | 18.7 |

*Table 5.* The Galileo models excel at image segmentation measured by % mIoU via linear probing (Galileo-Base obtains an average rank of 2.7, Table 18). We show the best architecture per method. We **bold** and underline the 1st and 2nd best results across all methods and architectures, as reported in Table 17. The Sen1Floods11 dataset consists of labelling floods from SAR data; models which do not support this modality have the result replaced with N/A.

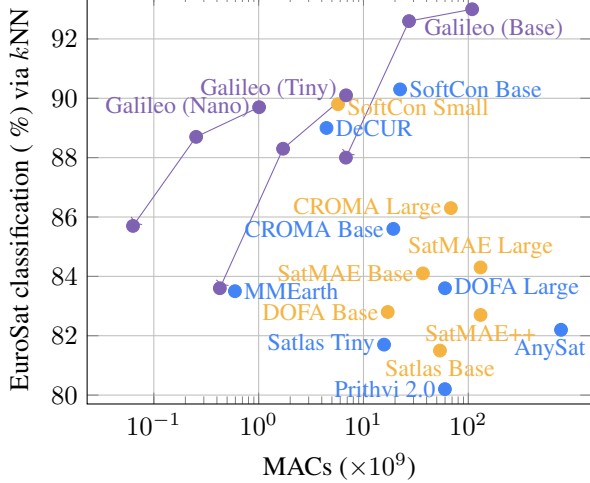

*Figure 4.* Accuracy trades off with inference cost as patch size varies in $\{4, 8, 16\}$. Cost is measured as Multiply-Accumulate operations (MACs) for encoding one EuroSat input (note the log scale on the x-axis). Galileo model size and patch size offer balance between performance and cost. See Table 14 for full results.

*Table 6.* The Galileo models are the best (-Base) and second-best (-Tiny) models for pixel timeseries classification, measured via linear probing. The best result is **bolded** and the second best is underlined. The CropHarvest dataset contains a number of modalities in addition to Sentinel-2 optical imagery, including topography, weather and SAR data. We use all modalities each model can support.

| Method | Arch. | CropHarvest Togo | CropHarvest Brazil | CropHarvest Kenya | CropHarvest Breizhcrops |
|---|---|---|---|---|---|
| Presto | ViT-Presto | **75.5** | 98.8 | 84.0 | 63.0 |
| AnySat | ViT-Base | 73.4 | 76.7 | 75.5 | 66.1 |
| **Galileo** | ViT-Nano | 73.5 | 76.4 | 84.5 | 67.3 |
| **Galileo** | ViT-Tiny | 74.7 | 97.2 | **85.4** | 69.0 |
| **Galileo** | ViT-Base | 74.8 | **99.3** | 84.2 | **73.0** |

AnySat and the pixel time series specialist Presto (Tab. 6). Galileo outranks Presto and far exceeds AnySat.

*Table 7.* Deep targets combined with structured space-time masking excels in **global** feature extraction. Segmentation tasks are gray-ed to focus on classification with our global task. We measure % top-1 accuracy via *k*NN.

| masking strategy | target enc. computation | loss function | MADOS | Floods | CropH. | EuroSat |
|---|---|---|---|---|---|---|
| space+time | varied | PatchDisc$_B$ | 58.91 | 76.92 | 88.72 | 89.50 |
| random | varied | PatchDisc$_B$ | 11.71 | 69.62 | 82.12 | 17.40 |
| random+space+time | varied | PatchDisc$_B$ | 22.87 | 71.62 | 76.53 | 66.30 |
| space+time | 0 | PatchDisc$_B$ | 61.73 | 76.66 | 85.79 | 86.90 |
| space+time | 6 | PatchDisc$_B$ | 63.83 | 76.93 | 88.17 | 89.20 |
| space+time | 12 | PatchDisc$_B$ | 60.35 | 77.19 | 87.30 | 87.90 |
| space+time | varied | MSE | 62.35 | 76.78 | 86.02 | 87.20 |
| space+time | varied | PatchDisc | 25.74 | 71.68 | 75.30 | 62.50 |

*Table 8.* Deep-shallow contrastive learning combined with unstructured random masking excels in **local** feature extraction. Classification tasks are gray-ed to focus on segmentation with our local task. We measure % mIoU (↑) of linear prediction on frozen features.

| masking strategy | target enc. computation | loss function | MADOS | Floods | CropH. | EuroSat |
|---|---|---|---|---|---|---|
| random | 0 | PatchDisc | 71.48 | 77.39 | 86.77 | 86.90 |
| random+space+time | 0 | PatchDisc | 68.63 | 77.82 | 85.31 | 88.80 |
| space+time | 0 | PatchDisc | 62.25 | 77.22 | 86.82 | 87.00 |
| random | 6 | PatchDisc | 58.53 | 75.66 | 76.58 | 65.40 |
| random | 12 | PatchDisc | 11.65 | 72.60 | 71.92 | 27.50 |
| random | varied | PatchDisc | 8.25 | 68.89 | 77.83 | 18.40 |
| random | 0 | MSE | 65.34 | 77.09 | 86.71 | 87.40 |
| random | 0 | PatchDisc$_B$ | 70.12 | 77.26 | 85.27 | 88.20 |

## 4.1. Ablations

For all our ablation experiments, we pretrain ViT-Tiny models for 200 epochs. We select four diverse tasks for segmentation (Sen1Floods11 and MADOS), image classification (EuroSat), and time series classification (CropHarvest), using only the validation sets for ablations.

We first ablate our global and local tasks: while the global task excels at the classification tasks and the local task excels at the segmentation tasks, neither excel at both. We then ablate our combined algorithm, which excels on both the classification and segmentation tasks. We ablate the following specific components of our algorithms:

**Global task ablations.** We focus on classification, which

*Table 9.* Our dual-objective algorithm excels on both classification and segmentation, and is more consistent than our single-objective algorithms. MADOS and Sen1Floods11 (% mIoU) via linear probing. CropHarvest and EuroSat (% top-1 acc.) via $k$NN.

| global loss | local loss | share predictors | target context | MADOS | Floods | CropH. | EuroSat |
|---|---|---|---|---|---|---|---|
| PatchDisc$_B$ | PatchDisc | no | all | 64.37 | 77.33 | 87.72 | 89.70 |
| PatchDisc | PatchDisc | no | all | 67.79 | 77.66 | 87.87 | 91.00 |
| PatchDisc$_B$ | PatchDisc | no | dec. | 63.54 | 76.95 | 86.98 | 89.30 |
| PatchDisc | PatchDisc | no | dec. | 36.98 | 74.21 | 85.49 | 83.30 |
| PatchDisc | PatchDisc | no | dec.+enc. | 63.41 | 77.36 | 85.87 | 89.30 |
| PatchDisc | PatchDisc | yes | all | 67.04 | 78.23 | 85.23 | 88.50 |
| PatchDisc$_B$ | PatchDisc$_B$ | no | all | 67.88 | 77.08 | 86.61 | 89.50 |
| MSE | MSE | no | all | 62.36 | 77.17 | 86.28 | 88.70 |

our global learning is designed for (see Tab. 7). Our global task chooses per-modality depths when computing targets. It slightly outperforms models that set all target depths to 6 (half the layers) and 12 (all layers). Using only linear projections for target processing drops by 2.6% on EuroSat and 2.9% on CropHarvest, confirming the importance of targeting deeper features for classification. Using the PatchDisc loss without our local task fails, and achieves only 62.5% on EuroSat, which may potentially be caused by a shortcut exploiting position embeddings. We fix this by including tokens from other samples in the batch as negatives in the contrastive objective (our PatchDisc$_B$). Finally, unstructured random masking fails when used in our global task.

**Local task ablations.** We focus on segmentation, which our local learning is designed for (see Tab. 8). Our local targets have a depth of 0, i.e., they are shallow linear projections of the input without any deeper modeling. The choice of shallow targets is highly effective: it achieves 71.5% mIoU on the MADOS dataset, which contains tiny objects such as marine debris, while our global task achieves only 58.9%. Using the PatchDisc loss slightly outperforms PatchDisc$_B$; only targeting linear projections (i.e., without position embeddings) prevents potential shortcuts without using negative tokens from the batch. These contrastive losses outperform the MSE loss by 5+% on MADOS, which demonstrates repelling the pixels from other tokens amplifies local features. Ours is the first use of deep-shallow contrastive learning for self-supervised learning for RS in particular and SSL in general. Finally, unstructured random masking outperforms structured masking by 9% on MADOS, which confirms our intuition that prediction across shorter spans promotes local features.

**Full algorithm ablations.** Although PatchDisc$_B$ is essential for our global task when used alone, when used with our local task it is unnecessary. Not sharing predictor parameters across objectives is optimal. Interestingly, our dual-objective strategy achieves successful training runs more consistently (e.g. 100% of runs achieve >80% on EuroSat in Tab. 9 vs. 63% of runs in Tabs. 7 and 8).

## 5. Related Work and Background

**Self-Supervised Learning.** Reconstructing a masked or noisy input is a common form of self-supervised pretraining, both for natural language (Devlin et al., 2018; Radford et al., 2018; Mikolov et al., 2013) and natural imagery (Xie et al., 2022; He et al., 2022; Vincent et al., 2008). While these methods make predictions in the input space (e.g. reconstructing pixels as done by the MAE He et al. (2022)), recent work has investigated making predictions in the latent space (Assran et al., 2022; Wei et al., 2024). These methods predict patch *representations* computed by the encoder's exponential moving average. Galileo uniquely predics and learns at *different depths* of the latent space, ranging from (linear projections of) the input space to the full depth of the latent space.

Contrastive learning (Le-Khac et al., 2020; Oord et al., 2018; Chen et al., 2020; Chopra et al., 2005) is a different approach to self-supervised pre-training: it duplicates and transforms inputs, encodes them all, then attracts the representations of the same input (called positives) and repels the representations of different inputs (called negatives). LatentMIM (Wei et al., 2024) applies contrastive learning to latent representations of masked inputs to increase the stability of these methods compared to reconstructive losses: its PatchDisc loss attracts patch representations of the same location within an image, and repels patch representations in the same input but at different locations. We adopt the PatchDisc loss but observe it remains prone to collapse. Galileo's *dual losses* stabilize pretraining for reliable improvement of the loss.

**Pretrained RS Models.** When pretraining models for remote sensing data, most methods process a *single timestep* of data, either of multispectral optical imagery only (Sat-MAE (Cong et al., 2022), MMEarth (Nedungadi et al., 2024)), multispectral optical imagery and SAR data seperately (SoftCon Wang et al. (2024b), DOFA Xiong et al. (2024), DeCUR Wang et al. (2024a)) or multispectral optical imagery and SAR data jointly (CROMA (Fuller et al., 2024)). Models which process multiple timesteps of data can either only process multispectral optical imagery (Prithvi 2.0 (Szwarcman et al., 2024), Satlas (Bastani et al., 2023)) or ignore spatial relationships and treat the data as pixel time series (Presto (Tseng et al., 2023)). These models employ different self-supervised learning methods during pretraining; we illustrate some of them in Figure 5.

Galileo learns from and processes more modalities than these previous approaches. It can jointly process multispectral optical imagery and SAR imagery *in addition* to many other remote sensing products, including topography, weather, population maps, night-lights and land cover classification maps. These products are commonly used in remote sensing tasks, and are therefore important for the utility of Galileo in a wide range of remote sensing applications.

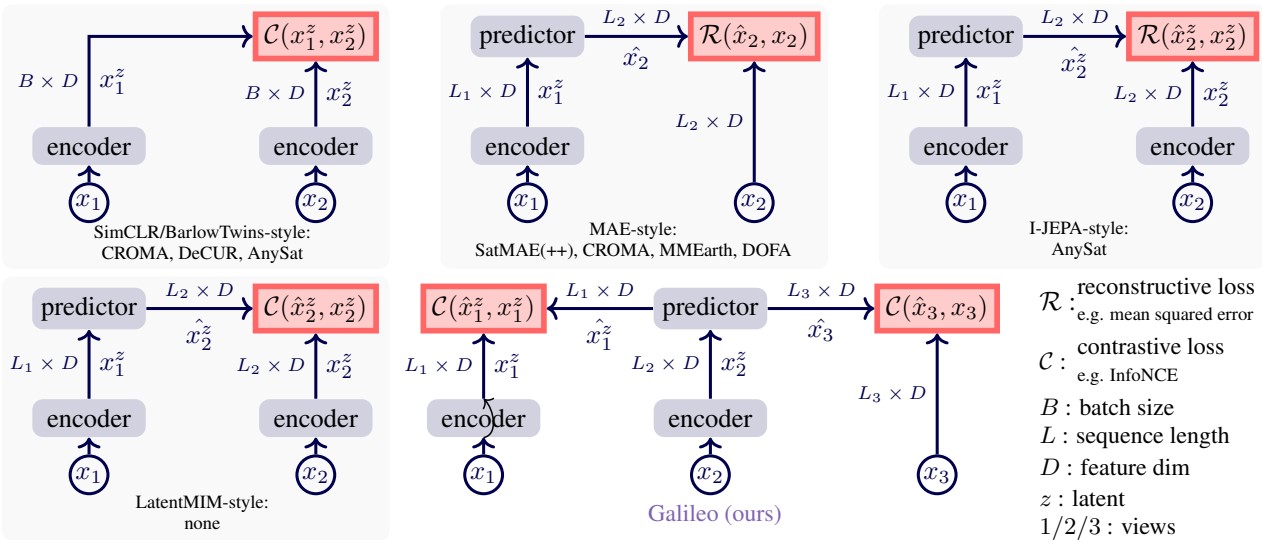

Figure 5. **SSL for RS**. **Top left:** Contrastive methods attract representations originating from the same sample and repel representations from other samples. **Top center:** Reconstruction methods predict pixels of hidden patches. **Top right:** I-JEPA and AnySat predict representations of hidden patches. **Bottom left:** LatentMIM (which lacks an RS instantiation) attracts representations originating from the same patch and repels representations from other patches. **Galileo (ours):** Our method simultaneously attracts varied-level representations of the same patch and repels elsewhere while it attracts pixel predictions of the same patch and repels elsewhere. This strategy encourages learning global *and* local features.

In addition, Galileo can *flexibly* model both the space and time dimensions of this multimodal data to handle inputs as single-timestep imagery, multi-timestep imagery, or pixel time series. This reflects the many multimodal, multi-shape approaches used by remote sensing practitioners (from pixel time series (Van Tricht et al., 2023; Kruse et al., 2023) to single- or multi- timestep imagery (Beukema et al., 2023)), and allows Galileo to fit seamlessly into existing remote sensing workflows.

AnySat (Astruc et al., 2024) is concurrent with our work and shares the same spirit. AnySat processes data from many satellites, and can also flexibly process the space and time dimensions of this data. However, AnySat is missing many of the other modalities processed by Galileo, which may be necessary to model a range of remote sensing phenomena (Poggio et al., 2021; Van Tricht et al., 2023)) and make an empirical difference across our benchmarks (Table 11).

## 6. Conclusion

We identify two requirements for the application of pretrained models in a wide range of RS contexts: (i) the ability to flexibly process different modalities and input shapes, and (ii) the ability to model RS phenomena which occur at different scales. To meet these requirements, we present the Galileo family of pretrained RS models.

We achieve these requirements by innovating on (i) the Galileo model architecture to flexibly process highly mul-

timodal inputs that vary in both space and time, (ii) our dual local-global SSL algorithm, to encourage the model to learn phenomena occurring at different scales, and (iii) the pretraining dataset used to train the Galileo models.

We run hundreds of evaluations — including extensive sweeps of baseline pretrained RS models — to robustly demonstrate Galileo's performance across a wide range of domains and task types. We run thorough ablations of our method. Having confirmed the effectiveness and transferability of unified local, global, and multimodal self-supervised learning with Galileo, we note that more research is needed to investigate local and global self-supervision for other data beyond RS.

The ability to process many remote sensing modalities is important to remote sensing practitioners, who find that using a range of inputs is critical to obtaining good results (Poggio et al., 2021; Rao et al., 2020; Van Tricht et al., 2023). This is rarely reflected in benchmark datasets, which typically only consist of optical or radar data. While many pretrained models can *only* process the benchmark modalities, Galileo is trained to process additional modalities that are common in practice. This functionality, despite not being captured by these standard benchmarks, is valuable to practitioners who need to take full advantage of the available data.

The model weights, pretraining code, pretraining data and evaluation code are open sourced at github.com/nasaharvest/galileo.

## Impact Statement

Applications of machine learning to RS span a range of societally important applications, from species distribution modelling (Teng et al., 2024) to disaster management (Kansakar & Hossain, 2016). By providing a set of RS models which can perform well even when few labels are available, we hope to enable RS practitioners to continue exploring and deploying these applications. We take several steps to encourage the adoption of these models, including training the models on publicly available RS data and training a diversity of model sizes so that they can be used in compute-constrained environments. In addition, we demonstrate Galileo's performance using both computationally expensive transfer learning (with finetuning) and computationally cheap transfer learning (with $k$NN and linear probing).

Tuia et al. (2023) highlight that a risk of these models is that they can be used to collect information about populations so that decisions are made without their involvement. We encourage the deployment of Galileo in collaboration with local communities and stakeholders (Krafft, 2023; Kshirsagar et al., 2021; Nakalembe & Kerner, 2023).

## Acknowledgments

AF is primarily supported by an NSERC PGS-D. DR and ES are supported by Canada CIFAR AI Chairs. GT is supported by the NSERC-CREATE LEADS program. GT and HK are supported by the NASA Harvest Grant #80NSSC23M0032.

This research was enabled in part by compute resources provided by Mila (mila.quebec), including material support from NVIDIA Corporation. We thank the Beaker team at Ai2 (Guerquin, 2022) for their support on Ai2's cluster.

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

# A. Methodology details

## A.1. The Galileo SSL algorithm

We adopt a latent prediction framework inspired by Assran et al. (2023), Garrido et al. (2024), and Wei et al. (2024), which operates as follows: ❶ Given a batch of samples, we construct two *different* views of each sample, $\mathbf{x}_1 \in \mathbb{R}^{L_1 \times D}$ and $\mathbf{x}_2 \in \mathbb{R}^{L_2 \times D}$. ❷ Our "online" encoder computes patch encodings $\mathbf{z}_1 = \mathbf{E}(\mathbf{x}_1)$, while our "target" encoder — an exponential moving average of the online encoder — computes target patch encodings $\mathbf{z}_2 = \mathbf{E}_{\text{EMA}}(\mathbf{x}_2)$. ❸ A predictor transformer $\mathbf{P}$ receives the target view's position, time, month, and channel group embeddings $\mathbf{e}_2 \in \mathbb{R}^{L_2 \times D}$ as placeholder queries and predicts patch encodings $\mathbf{p} \in \mathbb{R}^{L_2 \times D}$ by cross-attending to the online patch encodings, i.e., $\mathbf{p} = \mathbf{P}(\mathbf{e}_2, \mathbf{z}_1)$. ❹ The predictions $\mathbf{p}$ and targets $\mathbf{z}_2$ are compared to compute a loss $\mathcal{L}(\mathbf{p}, \mathbf{z}_2)$ that updates the online encoder.

*Table 10.* Our *global* task, with a varying minibatch size. By default, we use the largest minibach size which fits in memory on a single H100 GPU (32). We include PatchDisc$_I$ results, which are equivalent to using PatchDisc$_B$ with a minibatch size of 1.

| Minibatch size | MADOS | Floods | CropH. | EuroSat |
|---|---|---|---|---|
| 32 | 58.91 | 76.92 | 88.72 | 89.50 |
| 16 | 64.24 | 77.76 | 88.51 | 90.10 |
| 1 (PatchDisc$_I$) | 25.74 | 71.68 | 75.30 | 62.50 |

We adapt this latent prediction framework for learning local and global features. We outline those adaptations below.

### A.1.1. Learning Global Features

We design this algorithm to learn abstract, lower-frequency features suited for classification applications. ❶ View construction involves: (a) uniformly sampling the number of channel groups $N \in \{2, 3, \ldots, 17\}$, (b) randomly selecting $N$ channel groups (e.g., RGB, SAR, ERA5), (c) repeating steps (a-b) for the target encoder while excluding overlapping channel groups, (d) applying either spatial or temporal masking, and (e) tokenizing both views to obtain $\mathbf{x}_1$ and $\mathbf{x}_2$. Space masking samples masks across space while maintaining consistency across time; time masking does the same across time while maintaining consistency across space. In both cases, modalities are selected for encoding or decoding. ❷-❸ Following our general framework, we compute $\mathbf{z}_1$ and $\mathbf{p}$, and compute targets using varied exit depths from the target encoder, $\mathbf{E}_{\text{EMA}}^{\ell}$. ❹ To encourage globally discriminative representations, we extend the PatchDisc loss to better discriminate samples in a batch by sampling negative examples from the batch (as opposed to only the instance). This approach ("PatchDisc$_B$") is defined below (note this is equivalent to PatchDisc$_I$ if the $\sum_{i'}^{B}$ summation is removed):

$$\mathcal{L}(\mathbf{u}, \mathbf{v}) = \frac{-\tau}{B} \sum_i^B \frac{1}{L_i} \sum_j^{L_i} \log \frac{\exp(\text{sim}(\mathbf{u}_{i,j}, \mathbf{v}_{i,j})/\tau)}{\sum_{i'}^B \sum_{j'}^{L_{i'}} \exp(\text{sim}(\mathbf{u}_{i,j}, \mathbf{v}_{i',j'})/\tau)}$$

with the softmax temperature $\tau$, the sample index $i$, the batch size $B$, the token index $j$, the number of tokens in the $i^{th}$ sample $L_i$, and the $l_2$ normalized dot product $\text{sim}(\mathbf{u}, \mathbf{v}) = \mathbf{u}^\top \mathbf{v}/\|\mathbf{u}\|\|\mathbf{v}\|$. This yields the following global task:

$$\mathcal{L}_{global} = \text{PatchDisc}_B(\mathbf{P}(\mathbf{e}_2, \mathbf{E}(\mathbf{x}_1)), \text{sg}(\mathbf{E}_{\text{EMA}}^{\ell}(\mathbf{x}_2)))$$

**Measuring the effect of batch size**  PatchDisc$_B$ samples negative patches from all instances within a *batch*, compared to within an *instance* for PatchDisc$_I$. This introduces a dependency on batch size; we measure the effect of batch size in Table 10.

### A.1.2. Learning Local Features

We design this algorithm to learn fine-grained, higher-frequency features suited for segmentation applications. ❶ View construction involves: (a) tokenizing the entire sample, and (b) randomly selecting 5% of tokens for $\mathbf{x}_1$ and 50% for $\mathbf{x}_2$. ❷-❸ Following our general framework, we compute $\mathbf{z}_1$ and $\mathbf{p}$, but compute targets using only the target encoder's linear projection, i.e., $\mathbf{E}_{\text{EMA}}^{proj}$ — skipping transformer blocks such that the predictor targets low-level features. ❹ We use LatentMIM's PatchDisc loss, tasking the model to discriminate between patches on the basis of low-level features alone:

$$\mathcal{L}_{local} = \text{PatchDisc}(\mathbf{P}(\mathbf{e}_2, \mathbf{E}(\mathbf{x}_1)), \text{sg}(\mathbf{E}_{\text{EMA}}^{proj}(\mathbf{x}_2)))$$

### A.1.3. Combining Local and Global Objectives

As noted in Section 2.2, our combined method alternates between the local and global objectives during pretraining:

$$\mathcal{L}_{Galileo} = \tfrac{1}{2}(\mathcal{L}_{global} + \mathcal{L}_{local})$$

## B. Pretraining details

### B.1. A globally sampled pretraining dataset

To construct the Galileo dataset, we split the global WorldCover map (Zanaga et al., 2022) into $1000 \times 1000$ pixels ($10km \times 10km$) tiles. For each tile, we compute two feature sets: ❶ the number of pixels within each WorldCover classification class, and ❷ the latitude and longitude of the tile. We use these features to train a $k$=150,000 $k$-means clustering algorithm, and select the tiles closest to the centroid of each cluster. This yields 150,000 training points, of which 85% (127,155) are successfully exported using Google Earth Engine (Gorelick et al., 2017). By including both the pixel counts and the latitude and longitudes as features to the $k$-means algorithm, we ensure both the semantic and geographic diversity of the model's training points — Figure 3 shows a chloropleth map of the exported points.

We use this sampling procedure to construct a rich dataset to pretrain our model. This dataset consists of 9 RS inputs, ranging from directly sensed inputs (such as Sentinel-2 optical imagery) to semantically dense maps (such as the Dynamic World landcover maps) — these are discussed in detail in Section 2.3. Table 11 studies the impact of each of these modalities on the model's downstream performance, by pretraining the combined global-local model while omitting a single data product.

*Table 11.* Ablating the Galileo dataset. MADOS and Sen1Floods11 (% mIoU) via linear probing. CropHarvest and EuroSat (% OA) via $k$NN.

| Removed input | MADOS | Sen1Floods11 | CropHarvest | EuroSat |
|---|---|---|---|---|
| None | 67.79 | 77.66 | 87.87 | 91.00 |
| S1 | 67.67 | N/A | 85.27 | 90.20 |
| NDVI | 67.89 | 78.10 | 88.32 | 90.00 |
| ERA5 | 68.10 | 77.10 | 87.14 | 91.20 |
| TerraClim | 61.30 | 74.90 | 82.78 | 81.20 |
| VIIRS | 63.48 | 74.52 | 84.10 | 81.10 |
| SRTM | 66.14 | 77.62 | 86.74 | 91.00 |
| DynamicWorld | 67.24 | 77.86 | 87.80 | 89.30 |
| WorldCereal | 65.94 | 77.56 | 87.71 | 89.60 |
| LandScan | 60.74 | 77.45 | 87.89 | 91.10 |
| Location | 69.25 | 77.36 | 87.14 | 91.20 |

### B.2. Implementation

All models are trained on single H100 GPUs (model sizes and training times are described in Table 12). We use an effective batch size of 512, which consists of minibatches of 32 instances augmented and repeated 4 times (Hoffer et al., 2019). For data augmentations, we randomly apply vertical and horizontal flipping and 90-degree rotations to each instance. When repeating the data, we first randomly select a patch size $P \in [1, 2, 3, 4, 5, 6, 7, 8]$. We then randomly select a (size, timestep) combination $(S, T) \in [(4, 12), (5, 6), (6, 4), (7, 3), (9, 3), (12, 3)]$. We then randomly subset spatially height $H = P \times S$, width $W = P \times S$ and timesteps $T$ from each instance in the batch.

We use bfloat16 precision, and the AdamW optimizer with $\beta_1 = 0.9$ and $\beta_2 = 0.999$ with gradient clipping. We warmup our learning rate for 30 epochs to a maximum learning rate before applying a cooldown via a cosine decay schedule. We use exponential moving averaging (EMA) to update our target encoder with a momentum value of 0.996 which linearly increases to 1 throughout pretraining following Assran et al. (2022).

For all ablations (Section 4.1), we pretrain a ViT-Tiny model for 200 epochs to a maximum learning rate of $2 \times 10^{-3}$ and use a weight decay of 0.02. For the final Galileo models, we pretrain the models for 500 epochs and conduct a sweep of [learning rate × weight decay]. For the ViT-Nano and ViT-Tiny architectures, we sweep learning rates $\in [1 \times 10^{-3}, 2 \times 10^{-3}, 3 \times 10^{-3}]$ and weight decays $\in [1 \times 10^{-2}, 2 \times 10^{-2}, 3 \times 10^{-2}]$. For the ViT-Base architecture, we sweep learning rates $\in [1 \times 10^{-4}, 3 \times 10^{-4}, 1 \times 10^{-3}, 2 \times 10^{-3}, 3 \times 10^{-3}]$ and weight decays $\in [1 \times 10^{-2}, 2 \times 10^{-2}, 3 \times 10^{-2}]$.

*Table 12.* Configurations of our ViT models and associated pretraining costs. GPU-hours describes the number of GPU-hours required to pretrain each model for 500 epochs on an H100 GPU.

| architecture | blocks | dim | heads | params | GPU-hours |
|---|---|---|---|---|---|
| ViT-Nano | 4 | 128 | 8 | 0.8M | 200 |
| ViT-Tiny | 12 | 192 | 3 | 5.3M | 259 |
| ViT-Base | 12 | 768 | 12 | 85.0M | 573 |

# C. Evaluation details

## C.1. Implementation

To ensure consistent experimental settings when comparing pretrained models, we rerun all evaluations under identical conditions. For the $k$NN probing, we follow the implementation of Gwilliam & Shrivastava (2022) — we use the pretrained models to compute representations of the test data (as values) and training data (as keys) — we then use the keys to classify the test data. Following Fuller et al. (2024) and Reed et al. (2023), we use $k = 20$. When linear probing, we use the pretrained models to compute representations of the training data and use this to train linear probes. We sweep learning rates when training the linear probes ($\{1, 3, 4, 5\} \times 10^{\{-4, -3, -2, -1\}}$) and apply the trained linear probes to the computed representations of the test data. When finetuning, we sweep learning rates when finetuning ($\{1, 3, 6\} \times 10^{\{-5, -4, -3\}}$) and apply the finetuned models to the test data.

## C.2. Evaluation Datasets

We evaluate our models on the datasets described below. For all GeoBench-modified datasets (Lacoste et al., 2024) - m-Eurosat, m-BigEarthnet, m-So2Sat, m-Brick-Kiln, m-Cashew-Plant and m-SA-Crop-Type, we use the training, validation and test splits shared by GeoBench. In addition, we use the 1%, 5% and 20% partitions shared by GeoBench.

- **m-EuroSat** (Helber et al., 2019): The full training set consists of 2,000 images, with 1,000 images in the validation and test sets. Images are $64 \times 64$ pixels.

- **m-BigEarthNet** (Sumbul et al., 2019): The full training set consists of 20,000 images, with 1,000 images in the test set. Images are $120 \times 120$ pixels.

- **m-So2Sat** (Zhu et al., 2020): The full training set consists of 19,992 images, with 986 images in the test set, and images are $32 \times 32$ pixels.

- **m-Brick-Kiln** (Lee et al., 2021): The full training set consists of 15,063 images, with 999 images in the test set. Images are $64 \times 64$ pixels.

- **m-Cashew-Plant** (Yin et al., 2023): The full training set consists of 1,350 images, with 50 images in the test set. Images are $256 \times 256$; we subtile them into $64 \times 64$ images.

- **m-SA-crop-type** (link): The full training set consists of 3,000 images, with 93 images in the test set. Images are $256 \times 256$; we subtile them into $64 \times 64$ images.

- **MADOS** (Kikaki et al., 2024): The full MADOS dataset consists of 2,804 $140 \times 140$ images, extracted from 174 Sentinel-2 scenes. We use the train/val/test splits from MADOS (50%/25%/25%) — each split was created as a representative subset of the entire MADOS dataset. In addition, we subtile each image into $80 \times 80$ images.

- **PASTIS** (Garnot & Landrieu, 2021): The full PASTIS dataset consists of 2,433 $128 \times 128$ time series, with 38-61 timesteps per time series. We subtile each time series spatially into $64 \times 64$ images. In addition, we compute monthly aggregations of the time series. Garnot & Landrieu (2021) share 5 folds of the data; we use folds $\{1, 2, 3\}$ for training, 4 for validation and 5 for testing. When applying single-timestep models to this dataset, we additionally sweep pooling methods to pool per-timestep encodings (as described in Section C).

- **Breizhcrops** (Rußwurm et al., 2019): The Breizhcrops dataset consists of pixel time series in 4 NUTS-3 regions in Brittany, France. We use 2 for training (FRH01, with 178,613 parcels and FRH02 with 140,645 parcels). We use FRH03 (166,391 parcels) for validation and FRH04 (122,614 parcels) for testing. The dataset consists of variable sequence lengths; we compute monthly aggregations of the time series.

- **CropHarvest** (Tseng et al., 2021): The CropHarvest dataset consists of 3 pixel time series tasks: (i) crop vs. non crop in Togo, with 1,319 samples in the training set and 306 samples in the test set, (ii) maize vs. rest in Kenya with 1,345 samples in the training set and 1,942 $m^2$ of densely labelled pixels in the test set, and (iii) coffee vs. rest in Brazil with 794 samples in the training set and 4.2 $km^2$ of densely lablled pixels in the test set.

## C.3. Comparing to baseline models

Corley et al. (2024) found that input-image sizes and feature scaling methods can have significant impacts on the performance of pretrained RS models. We therefore resize all input images to the sizes that the models were pretrained on. In addition, we treat feature scaling methods as an additional hyperparameter, and sweep it in addition to the learning rates (where those are applicable, i.e. for linear probing and finetuning). Finally, the PASTIS dataset consists of multiple timesteps of optical imagery. Since all benchmark models (except AnySat) cannot process the full time series natively, we use multiple forward passes. We test a mean and a max to combine the model outputs, following Bastani et al. (2023).

The reported test results are therefore computed by sweeping the cross product of the following hyperparameters, using the validation sets in the downstream datasets:

$$[\text{Learning Rate}] \times [\text{Temporal aggregations}]$$

In addition to conducting this sweep, we run the linear probes 5 times and average the results. When running the linear probe, we sweep the learning rate and feature scaling method concurrently for the first run. We select the feature scaling method from this first run, and fix it for all subsequent runs. We then select the best other hyperparameters per run, and aggregate these to obtain our final results.

*Table 13.* Galileo MADOS classification test performance (%) as a function of patch size measured via linear probing for different training set %s.

| Arch. | patch size | 100 % | 20 % | 5% | 1% |
|---|---|---|---|---|---|
| ViT-Nano | 2 | 53.6 | 43.9 | 33.5 | 16.6 |
| | 4 | 54.8 | 41.5 | 28.9 | 13.9 |
| ViT-Tiny | 2 | 61.9 | 49.9 | 32.6 | 15.2 |
| | 4 | 60.8 | 50.6 | 34.0 | 17.5 |
| ViT-Base | 2 | 68.4 | 53.4 | 39.0 | 18.0 |
| | 4 | 67.6 | 49.0 | 34.1 | 14.7 |

We run this sweep for all evaluation datasets with the exception of the CropHarvest tasks; these consist of small training sets and no validation sets against which the hyperparameters can be selected. We therefore follow Tseng et al. (2023) in using the same feature scaling methods as was used during pretraining, and using scikit-learn's regression algorithm with default parameters (Pedregosa et al., 2011) for all models.

### C.3.1. FEATURE SCALING

The pretrained models we benchmark against apply either standardization (MMEarth, DOFA, AnySat and Presto) or normalization (all other models) during pretraining. We sweep the following normalization statistics, either via standardization on normalization depending on the pre-training procedure: ❶ statistics from the downstream datasets, ❷ SatMAE pretraining statistics, ❸ SSL4EO (Wang et al., 2023) statistics, ❹ Galileo pretraining dataset statistics, ❺ Presto pretraining dataset statistics. For all of these statistics, we additionally sweep standard deviation multipliers. Prithvi 2.0 statistics only cover a subset of Sentinel-2 bands; we therefore only include those statistics in the sweeps for the Prithvi 2.0 model.

## D. Results

We include full results for the image classification tasks (Table 15) and segmentation tasks (Table 17). In addition, full results for the m-Eurosat dataset with varying patch sizes are recorded in Table 14 - these values are used in Figure 4. Similarly, we measure results for MADOS with varying patch sizes in Table 13 - a patch size of 4 is used in Tables 5 and 17.

We rank the models in Table 18. When ranking the models, we compute the average rank of each model across each dataset and partition.

*Table 14.* Galileo m-Eurosat classification test performance (%) as a function of patch size measured via $k$NN for different training set %s. MACs required to process a single EuroSat instance are also recorded; by selecting the model size and patch size, practitioners can make trade offs between model performance and inference costs.

| Arch. | patch size | GMACs | 100 % | 20 % | 5% | 1% |
|---|---|---|---|---|---|---|
| ViT-Nano | 8 | 0.25 | 88.7 | 81.9 | 55.0 | 38.5 |
| | 16 | 0.06 | 85.7 | 79.3 | 56.0 | 41.1 |
| ViT-Tiny | 8 | 1.71 | 88.3 | 83.0 | 59.7 | 41.3 |
| | 16 | 0.43 | 83.6 | 78.4 | 50.1 | 33.8 |
| ViT-Base | 8 | 27.20 | 92.6 | 88.3 | 72.4 | 56.9 |
| | 16 | 6.80 | 88.0 | 82.4 | 58.6 | 48.9 |

*Table 15.* Image classification test performance (%) via *k*NN. Ranks are calculated by averaging all results and ranking the averages.

| Method | Arch. | m-EuroSat Training %, Top-1 Acc. ↑ | | | | m-BigEarthNet Training %, F1 Score ↑ | | | | m-So2Sat Training %, Top-1 Acc. ↑ | | | | m-Brick-Kiln Training %, Top-1 Acc. ↑ | | | |
|---|---|---|---|---|---|---|---|---|---|---|---|---|---|---|---|---|---|
| | | 100% | 20% | 5% | 1% | 100% | 20% | 5% | 1% | 100% | 20% | 5% | 1% | 100% | 20% | 5% | 1% |
| SatMAE (Cong et al., 2022) | ViT-Base | 84.1 | 73.3 | 50.1 | 34.8 | 50.6 | 42.5 | 35.7 | 29.0 | 36.0 | 32.9 | 29.7 | 23.1 | 86.1 | 81.9 | 80.3 | 73.5 |
| SatMAE (Cong et al., 2022) | ViT-Large | 84.3 | 74.7 | 53.1 | 46.4 | 50.8 | 42.9 | 35.6 | 27.7 | 36.6 | 34.3 | 31.0 | 24.4 | 87.9 | 84.0 | 80.4 | 74.7 |
| SatMAE++ (Noman et al., 2024) | ViT-Large | 82.7 | 75.9 | 51.1 | 48.5 | 50.8 | 42.8 | 36.7 | 31.6 | 34.7 | 32.7 | 29.9 | 23.4 | 89.6 | 87.1 | 82.8 | 76.7 |
| CROMA (Fuller et al., 2024) | ViT-Base | 85.6 | 79.4 | 66.2 | 51.3 | 58.8 | 55.3 | 49.3 | 44.7 | 48.8 | 48.0 | 43.9 | 33.8 | 92.6 | 90.6 | 87.7 | 85.1 |
| CROMA (Fuller et al., 2024) | ViT-Large | 86.3 | 78.1 | 59.9 | 49.0 | 56.6 | 50.6 | 44.1 | 38.0 | 47.6 | 45.0 | 43.2 | 33.7 | 91.0 | 86.7 | 82.9 | 80.2 |
| SoftCon (Wang et al., 2024b) | ViT-Small | 89.8 | 83.4 | 55.9 | 27.2 | 64.7 | 58.7 | 52.6 | 43.3 | 51.1 | 49.9 | 43.3 | 31.4 | 89.2 | 86.9 | 80.5 | 77.8 |
| SoftCon (Wang et al., 2024b) | ViT-Base | 90.3 | 82.1 | 54.2 | 19.8 | 63.7 | 57.5 | 52.0 | 42.5 | 51.0 | 49.7 | 45.3 | 35.4 | 90.0 | 86.1 | 80.6 | 74.5 |
| DOFA-v1 (Xiong et al., 2024) | ViT-Base | 82.8 | 72.1 | 60.9 | 49.6 | 49.4 | 43.6 | 37.2 | 29.9 | 41.4 | 40.7 | 37.5 | 29.4 | 88.3 | 86.2 | 82.0 | 78.3 |
| DOFA-v1 (Xiong et al., 2024) | ViT-Large | 83.6 | 72.1 | 53.5 | 41.7 | 49.9 | 41.6 | 35.3 | 27.6 | 45.4 | 40.6 | 35.6 | 31.8 | 86.8 | 85.2 | 84.8 | 80.6 |
| Satlas (Bastani et al., 2023) | Swin-Tiny | 81.7 | 70.3 | 48.3 | 35.8 | 51.9 | 44.8 | 37.8 | 29.6 | 36.6 | 30.7 | 29.6 | 27.1 | 88.2 | 85.2 | 82.4 | 73.0 |
| Satlas (Bastani et al., 2023) | Swin-Base | 81.5 | 69.1 | 42.1 | 10.0 | 47.0 | 41.1 | 35.0 | 25.8 | 35.8 | 33.4 | 29.6 | 30.4 | 80.0 | 78.3 | 76.9 | 73.3 |
| MMEarth (Nedungadi et al., 2024) | CNN-atto | 81.7 | 73.5 | 60.3 | 30.0 | 58.3 | 52.2 | 46.5 | 39.6 | 39.8 | 38.8 | 36.8 | 25.1 | 89.4 | 85.4 | 84.1 | 79.7 |
| DeCUR (Wang et al., 2024a) | ViT-Small | 89.0 | 85.3 | 72.3 | 46.6 | 63.8 | 59.2 | 55.4 | 49.6 | 45.8 | 43.1 | 38.5 | 30.9 | 83.7 | 81.7 | 77.9 | 74.2 |
| Prithvi 2.0 (Szwarcman et al., 2024) | ViT-Large | 80.2 | 69.4 | 54.1 | 48.0 | 49.4 | 42.9 | 35.5 | 28.8 | 29.5 | 31.2 | 29.6 | 26.1 | 87.9 | 86.8 | 83.3 | 80.6 |
| AnySat (Astruc et al., 2024) | ViT-Base | 82.2 | 73.7 | 62.5 | 47.1 | 54.9 | 47.2 | 40.7 | 33.7 | 39.8 | 34.9 | 32.0 | 29.0 | 85.3 | 81.7 | 78.0 | 72.0 |
| Galileo | ViT-Nano | 89.7 | 82.4 | 56.6 | 41.7 | 53.8 | 46.3 | 41.5 | 33.9 | 50.1 | 50.3 | 47.5 | 37.4 | 86.7 | 82.2 | 83.2 | 79.7 |
| Galileo | ViT-Tiny | 90.1 | 83.9 | 59.5 | 41.3 | 55.5 | 48.2 | 41.6 | 34.4 | 49.7 | 50.5 | 44.2 | 36.2 | 86.9 | 83.7 | 83.8 | 77.3 |
| Galileo | ViT-Base | 93.0 | 88.5 | 71.3 | 56.6 | 59.0 | 51.5 | 45.4 | 36.5 | 54.8 | 53.8 | 51.1 | 43.2 | 90.7 | 86.9 | 85.8 | 78.0 |

*Table 16.* Image classification test performance (%) via finetuning.

| Method | Arch. | m-EuroSat Training %, Top-1 Acc. ↑ | | | | m-BigEarthNet Training %, F1 Score ↑ | | | | m-So2Sat Training %, Top-1 Acc. ↑ | | | | m-Brick-Kiln Training %, Top-1 Acc. ↑ | | | |
|---|---|---|---|---|---|---|---|---|---|---|---|---|---|---|---|---|---|
| | | 100% | 20% | 5% | 1% | 100% | 20% | 5% | 1% | 100% | 20% | 5% | 1% | 100% | 20% | 5% | 1% |
| SatMAE (Cong et al., 2022) | ViT-Base | 96.5 | 90.8 | 79.7 | 55.5 | 67.8 | 59.3 | 51.1 | 39.0 | 54.5 | 52.0 | 45.2 | 34.8 | 98.5 | 97.4 | 97.0 | 94.0 |
| SatMAE (Cong et al., 2022) | ViT-Large | 96.6 | 91.5 | 82.5 | 56.9 | 68.3 | 61.1 | 52.4 | 41.8 | 57.2 | 56.2 | 49.7 | 36.4 | 98.4 | 97.3 | 97.3 | 96.1 |
| SatMAE++ (Noman et al., 2024) | ViT-Large | 96.5 | 90.6 | 80.1 | 56.4 | 67.9 | 60.4 | 51.9 | 45.6 | 56.0 | 52.4 | 46.0 | 36.9 | 98.6 | 97.3 | 96.0 | 92.5 |
| CROMA (Fuller et al., 2024) | ViT-Base | 96.0 | 91.2 | 79.2 | 53.6 | 70.0 | 63.4 | 54.0 | 43.4 | 59.7 | 59.1 | 54.1 | 43.3 | 98.7 | 97.8 | 97.0 | 96.1 |
| CROMA (Fuller et al., 2024) | ViT-Large | 96.6 | 92.9 | 80.7 | 52.7 | 71.9 | **66.0** | **58.3** | **47.9** | 60.6 | 57.9 | 52.9 | 40.9 | 98.7 | 98.0 | 97.1 | 96.7 |
| SoftCon (Wang et al., 2024b) | ViT-Small | 97.4 | 95.4 | 84.9 | 57.5 | 69.5 | 62.5 | 53.3 | 36.0 | 61.7 | 60.3 | 54.2 | 49.2 | 98.8 | **98.1** | 97.7 | 97.2 |
| SoftCon (Wang et al., 2024b) | ViT-Base | 97.5 | 95.0 | **88.2** | 56.3 | 70.3 | 63.6 | 53.8 | 38.5 | 61.7 | 60.3 | 54.2 | 49.2 | 98.7 | **98.1** | 98.0 | **97.3** |
| DOFA-v1-v1 (Xiong et al., 2024) | ViT-Base | 94.6 | 86.1 | 74.2 | 50.9 | 68.1 | 60.3 | 51.9 | 41.9 | 56.7 | 49.9 | 45.8 | 33.8 | 98.7 | 97.3 | 96.2 | 95.0 |
| DOFA-v1-v1 (Xiong et al., 2024) | ViT-Large | 96.9 | 91.5 | 82.2 | 53.4 | 68.0 | 60.3 | 52.2 | 43.5 | 58.7 | 55.4 | 47.4 | 37.0 | 98.6 | 96.9 | 96.1 | 94.5 |
| Satlas (Bastani et al., 2023) | Swin-Tiny | 96.3 | 89.1 | 78.1 | 52.9 | 71.3 | 63.8 | 53.6 | 32.0 | 57.3 | 52.7 | 45.9 | 30.8 | 98.5 | 97.7 | 96.8 | 94.7 |
| Satlas (Bastani et al., 2023) | Swin-Base | 97.5 | 92.2 | 81.2 | 51.9 | **72.8** | 65.1 | 54.9 | 25.8 | 61.9 | 55.0 | 47.0 | 30.6 | 98.4 | 97.9 | 97.2 | 94.7 |
| MMEarth (Nedungadi et al., 2024) | CNN-atto | 95.7 | 86.1 | 73.0 | 47.5 | 70.0 | 62.7 | 52.6 | 43.4 | 57.2 | 51.0 | 44.1 | 30.0 | **98.9** | 98.0 | 96.5 | 89.2 |
| DeCUR (Wang et al., 2024a) | ViT-Small | **97.9** | 95.3 | 87.9 | 54.2 | 70.9 | 64.9 | 54.7 | 44.7 | 61.7 | **61.0** | 54.2 | 47.0 | 98.7 | 98.0 | 97.1 | 96.9 |
| Prithvi 2.0 (Szwarcman et al., 2024) | ViT-Large | 96.5 | 89.2 | 77.6 | 51.5 | 69.0 | 61.8 | 51.4 | 37.1 | 54.6 | 50.5 | 40.2 | 31.0 | 98.6 | 97.6 | 96.7 | 96.2 |
| AnySat (Astruc et al., 2024) | ViT-Base | 95.9 | 88.2 | 74.4 | 51.3 | 70.3 | 61.6 | 46.1 | 13.3 | 51.8 | 49.8 | 42.0 | 29.7 | 98.6 | 97.2 | 96.8 | 85.6 |
| **Galileo (ours)** | ViT-Nano | 94.5 | 88.3 | 80.2 | 52.6 | 67.1 | 59.3 | 44.1 | 23.3 | 57.4 | 54.7 | 47.8 | 34.9 | 98.5 | 97.7 | 96.1 | 94.2 |
| **Galileo (ours)** | ViT-Tiny | 96.9 | 94.4 | 85.2 | 60.6 | 69.7 | 62.2 | 53.4 | 39.5 | 61.9 | 57.2 | 54.9 | 43.1 | 98.7 | 97.9 | 97.2 | 96.6 |
| **Galileo (ours)** | ViT-Base | 97.7 | **96.0** | 87.0 | **63.5** | 70.7 | 63.1 | 53.9 | 40.9 | **63.3** | 57.8 | **56.7** | **50.6** | 98.7 | 98.0 | 97.5 | 96.8 |

*Table 17.* Image (and image timeseries) segmentation test performance (%) via linear probing. * For semantic segmentation, AnySat outputs dense per-pixel features instead of per-patch. To keep the training-costs of the linear probes similar to other models, we sampled 6.25% of pixel features per image when training the linear probe for AnySat. Evaluation used all pixel features in an image.

| Method | Arch. | m-Cashew-Plant Training %, mIoU ↑ | | | | m-SA-Crop-Type Training %, mIoU ↑ | | | | MADOS Training %, mIoU ↑ | | | | Sen1Floods11 Training %, mIoU ↑ | | | | PASTIS Training %, mIoU ↑ | | | |
|---|---|---|---|---|---|---|---|---|---|---|---|---|---|---|---|---|---|---|---|---|---|
| | | 100% | 20% | 5% | 1% | 100% | 20% | 5% | 1% | 100% | 20% | 5% | 1% | 100% | 20% | 5% | 1% | 100% | 20% | 5% | 1% |
| SatMAE (Cong et al., 2022) | ViT-Base | 28.9 | 28.1 | 27.6 | 23.0 | 23.8 | 23.4 | 21.5 | 16.8 | 53.2 | 39.1 | 26.4 | 12.4 | not supported | | | | 27.6 | 24.2 | 18.5 | 11.2 |
| SatMAE (Cong et al., 2022) | ViT-Large | 30.8 | 29.7 | 28.7 | 22.7 | 24.8 | 24.0 | 21.9 | 16.9 | 55.6 | 41.0 | 29.9 | 13.2 | not supported | | | | 29.6 | 25.3 | 19.1 | 11.5 |
| SatMAE++ (Noman et al., 2024) | ViT-Large | 29.6 | 28.0 | 27.5 | 23.3 | 25.7 | 24.3 | 21.5 | 16.8 | 49.9 | 38.2 | 27.5 | 12.7 | not supported | | | | 30.5 | 26.0 | 19.3 | 12.0 |
| CROMA (Fuller et al., 2024) | ViT-Base | 31.8 | 31.4 | 30.2 | 26.8 | **32.0** | **29.9** | **26.1** | 18.3 | 64.2 | 49.1 | **39.6** | 24.4 | 78.9 | 78.1 | 77.4 | 77.6 | 44.4 | 38.4 | 29.2 | 18.5 |
| CROMA (Fuller et al., 2024) | ViT-Large | **34.3** | **33.3** | 32.5 | 27.9 | **32.0** | **29.9** | 25.6 | 18.0 | 66.3 | 52.5 | 36.2 | 13.9 | 78.6 | 78.0 | 77.1 | 77.2 | 42.9 | 35.9 | 25.8 | 16.1 |
| SoftCon (Wang et al., 2024b) | ViT-Small | 27.0 | 26.8 | 25.6 | 23.0 | 28.5 | 27.8 | 24.3 | 17.7 | 57.1 | 44.0 | 29.4 | 19.1 | 78.5 | 78.3 | 76.9 | 75.6 | 28.6 | 26.1 | 19.3 | 11.8 |
| SoftCon (Wang et al., 2024b) | ViT-Base | 29.6 | 28.9 | 27.2 | 22.8 | 30.8 | 29.3 | 24.7 | 18.5 | 60.3 | 42.4 | 31.9 | 16.5 | 78.0 | 77.4 | 74.9 | 74.8 | 31.3 | 26.5 | 19.3 | 10.5 |
| DOFA-v1 (Xiong et al., 2024) | ViT-Base | 26.9 | 26.7 | 26.8 | 22.2 | 24.8 | 23.9 | 21.0 | 16.6 | 48.3 | 37.4 | 30.0 | 19.1 | 78.1 | 77.8 | 77.0 | 77.1 | 29.8 | 25.6 | 19.5 | 13.2 |
| DOFA-v1 (Xiong et al., 2024) | ViT-Large | 27.7 | 27.4 | 27.3 | 23.3 | 25.4 | 23.9 | 21.3 | 16.8 | 51.6 | 38.5 | 31.0 | 19.1 | 78.1 | 77.9 | 77.3 | 77.4 | 29.8 | 25.5 | 19.5 | 13.4 |
| Satlas (Bastani et al., 2023) | Swin-Tiny | 25.1 | 24.8 | 24.2 | 18.6 | 23.4 | 22.7 | 19.8 | 16.2 | 45.9 | 35.7 | 26.5 | 12.4 | not supported | | | | 28.0 | 24.0 | 17.4 | 10.9 |
| Satlas (Bastani et al., 2023) | Swin-Base | 24.5 | 24.4 | 23.3 | 19.4 | 22.4 | 21.6 | 19.3 | 14.7 | 48.0 | 36.5 | 25.9 | 15.9 | not supported | | | | 25.4 | 21.6 | 16.1 | 9.2 |
| MMEarth (Nedungadi et al., 2024) | CNN-atto | 24.2 | 24.6 | 24.6 | 20.3 | 22.2 | 21.0 | 18.7 | 14.1 | 34.2 | 26.4 | 19.5 | 16.1 | not supported | | | | 24.0 | 21.6 | 16.0 | 10.5 |
| DeCUR (Wang et al., 2024a) | ViT-Small | 26.2 | 26.2 | 26.0 | 22.8 | 21.5 | 20.8 | 19.2 | 15.3 | 54.8 | 40.9 | 30.3 | 16.6 | 74.5 | 74.6 | 73.5 | 72.2 | 22.4 | 19.7 | 15.4 | 11.0 |
| Prithvi 2.0 (Szwarcman et al., 2024) | ViT-Large | 26.7 | 26.6 | 26.8 | 23.2 | 22.9 | 22.3 | 20.3 | 15.7 | 50.0 | 41.8 | 33.7 | 18.9 | not supported | | | | 29.3 | 26.8 | 20.2 | 13.2 |
| AnySat * (Astruc et al., 2024) | ViT-Base | 26.1 | 26.1 | 24.9 | 21.7 | 27.1 | 25.2 | 21.4 | 15.8 | 50.2 | 39.8 | 30.5 | 17.0 | 77.9 | 77.6 | 77.1 | 76.9 | **46.2** | **41.9** | **33.7** | **23.5** |
| Galileo | ViT-Nano | 24.4 | 24.6 | 24.6 | 24.5 | 19.7 | 19.7 | 17.1 | 14.5 | 54.8 | 41.4 | 28.9 | 13.9 | 78.6 | 78.5 | 77.7 | 77.1 | 17.5 | 17.0 | 15.7 | 13.1 |
| Galileo | ViT-Tiny | 27.4 | 27.0 | 27.3 | 27.9 | 22.5 | 22.4 | 20.5 | 17.1 | 60.8 | 50.6 | 34.0 | 17.5 | 78.0 | 77.8 | 77.7 | 77.9 | 28.1 | 27.0 | 23.1 | 16.9 |
| Galileo | ViT-Base | 33.0 | 32.8 | **33.1** | **30.2** | 30.1 | 29.3 | 25.4 | **19.4** | 67.6 | 49.0 | 34.1 | 14.7 | **79.4** | **79.0** | **78.5** | **78.2** | 39.2 | 36.7 | 27.9 | 18.7 |

*Table 18.* Model rankings, computed against the full Image Clasification (Im. Class.) results in Table 15, Image Segmentation (Im. Seg.) results in Table 17 and TimeSeries (TS) results in Table 6. We aggregate the Image Classification and Image Segmentation rankings into a single "Image" (Im.) rankings. When we do this, we average the rankings across all the tasks (as opposed to naively averaging the aggregated image classification and image segmentation rankings).

| Method | Arch. | Im. Class. | | Im. Seg | | |
| | | KNN | FT | LP | Im. | TS |
| --- | --- | --- | --- | --- | --- | --- |
| SatMAE (Cong et al., 2022) | ViT-Base | 13.8 | 12.5 | 11.7 | 12.6 | N/A |
| SatMAE (Cong et al., 2022) | ViT-Large | 11.9 | 9.1 | 10.1 | 10.4 | N/A |
| SatMAE++ (Noman et al., 2024) | ViT-Large | 10.9 | 11.4 | 10.4 | 10.9 | N/A |
| CROMA (Fuller et al., 2024) | ViT-Base | 3.6 | 7.4 | **2.5** | 4.3 | N/A |
| CROMA (Fuller et al., 2024) | ViT-Large | 5.9 | 5.3 | 3.5 | 4.8 | N/A |
| SoftCon (Wang et al., 2024b) | ViT-Small | 5.6 | 4.7 | 7.7 | 6.1 | N/A |
| SoftCon (Wang et al., 2024b) | ViT-Base | 5.9 | 4.0 | 7.3 | 5.9 | N/A |
| DOFA-v1 (Xiong et al., 2024) | ViT-Base | 9.4 | 13.1 | 9.6 | 10.6 | N/A |
| DOFA-v1 (Xiong et al., 2024) | ViT-Large | 10.6 | 10.2 | 7.7 | 9.4 | N/A |
| Satlas (Bastani et al., 2023) | Swin-Tiny | 12.7 | 10.6 | 14.9 | 12.9 | N/A |
| Satlas (Bastani et al., 2023) | Swin-Base | 15.9 | 7.9 | 15.7 | 13.4 | N/A |
| MMEarth (Nedungadi et al., 2024) | CNN-atto | 8.3 | 11.7 | 16.1 | 12.3 | N/A |
| DeCUR (Wang et al., 2024a) | ViT-Small | 7.0 | 3.6 | 13.0 | 8.3 | N/A |
| Prithvi 2.0 (Szwarcman et al., 2024) | ViT-Large | 12.0 | 12.5 | 10.8 | 11.7 | N/A |
| AnySat (Astruc et al., 2024) | ViT-Base | 11.1 | 14.5 | 8.3 | 11.1 | 4.5 |
| Presto (Tseng et al., 2023) | ViT-Presto | N/A | N/A | N/A | N/A | 3.0 |
| **Galileo** | ViT-Nano | 7.0 | 13.1 | 12.2 | 10.9 | 3.5 |
| **Galileo** | ViT-Tiny | 6.6 | 5.8 | 6.8 | 6.4 | 2.3 |
| **Galileo** | ViT-Base | **2.9** | **3.5** | 2.7 | **3.0** | **1.8** |

