# OpenReview forum: "Galileo: Learning Global & Local Features of Many Remote Sensing Modalities"
_ICML.cc/2025/Conference — ICML 2025 poster_

### Official Review · Reviewer_1X8T · 2025-03-10

**Overall Recommendation:** 3

**Summary:**

This work introduces a vision foundation model for remote sensing data based on self-supervised learning. The proposed method features two key technical designs: 1) A flexible encoder architecture that supports space-time, spatial, temporal, and static data. 2)A new training objectives that incorporate both global and local features to better capture representations for objects of varying scales and types. Extensive experiments conducted on multiple benchmark datasets demonstrate promising performance.

**Claims And Evidence:**

The claimed contributions are well-supported by the proposed method and experimental results.

**Essential References Not Discussed:**

No. The literature review looks comprehensive.

**Experimental Designs Or Analyses:**

The literature review appears comprehensive, and I am quite familiar with the topic.

**Methods And Evaluation Criteria:**

The method combines the strengths of contrastive learning and masked image modeling to learn features at both token and pixel levels. The approach is technically sound, and the evaluation metrics are reasonable.

**Other Comments Or Suggestions:**

No

**Other Strengths And Weaknesses:**

Strengths:
1) The idea of building a unified foundation model for diverse types of remote sensing data is compelling. Previous attempts have focused on unifying models from the perspective of spatial, temporal, or spectral characteristics. This work integrates several recent advances to create a more comprehensive and unified foundation model.

2) The experimental section is thorough and well-executed. Extensive experiments across multiple benchmarks effectively demonstrate the efficacy of the proposed method.

Weaknesses:
1) The proposed method combines training objectives from contrastive learning and masked image modeling with only minor modifications. While effective, the technical novelty is somewhat limited due to this straightforward and intuitive combination.

2) The technical distinction between PatchDisc and the proposed AllDisc is minimal.

3) The rationale behind the choice of PatchDisc and AllDisc is unclear. In Section 2.2.3, the authors state, "PatchDisc outperforms AllDisc when combining global and local objectives, so we use PatchDisc for both objectives." If this is the case, the necessity of introducing AllDisc in Section 2.2.1 becomes questionable. Additionally, Table 7 shows that the performance difference between the two is marginal.

4) The masking strategy is frequently mentioned in the experimental section but is less emphasized in the method description. If space-time masking consistently performs best, its discussion in the results section may be redundant. I also recommend removing discussions on target encoder exit depth in some tables, as they may distract readers from the main findings.

5) From Tables 3 and 4, it appears that the proposed method does not outperform CROMA on several benchmarks. Given that the proposed method is significantly more complex than CROMA and likely trained on a larger dataset, the performance gains do not seem substantial enough to justify the added complexity.

**Questions For Authors:**

See the weakness part.

**Relation To Broader Scientific Literature:**

Vision foundation models are of great interest to people in remote sensing and computer vision.

**Theoretical Claims:**

No theoretical proof in the paper.

---

> ### Author Rebuttal · Authors · 2025-03-31
>
> We thank the reviewer for the thoughtful feedback, for your attention to detail and for acknowledging the strengths of our submission.
>
> ### 1. Technical Novelty
> Our work introduces several technical innovations for self-supervised learning in general and pretraining models for remote sensing specifically:
>
> 1. We are the first to successfully fuse a diverse set of multiple sensors and products (e.g. weather) across space and time in a pretrained model. This fusion broadens Galileo's real-world applicability — as these inputs make a difference to downstream tasks [2,3,4] — and offers important empirical findings for multimodal self-supervision (e.g. the contributions of modalities to performance in Table 9).
> 2. We introduce contrastive learning with _varied depth_ targets, which we show is highly effective (8% improvement on the MADOS benchmark, Table 7). Our algorithms construct target tokens from (i) linear projections of the inputs (in our local loss) and (ii) a varying number of target encoder layers (in our global loss). Galileo is the first to exploit early-exited targets in SSL - we show its effectiveness via extensive ablations.
> 3. Galileo constitutes a novel combination of our dual local and global SSL losses which are each novel on their own. We demonstrate the efficacy of the algorithms in isolation (Tables 6, 7) and when combined (Table 8).
> 4. A novel combination of existing methods can also constitute technical novelty. For example, CROMA [NeurIPS ‘23] combined MAE and radar-optical contrastive learning: these SSL methods, _independently_, were not novel to remote sensing or ML in general. However, its _joint_ reconstructive and contrastive multi-modal self-supervisory method was novel.
> 5. We note that this work is submitted as “Application-Driven Machine Learning” as implemented by the [ICML 2025 process](https://icml.cc/Conferences/2025/ReviewerInstructions). This has different reviewing criteria including “Originality need not mean wholly novel method: It may mean a novel combination of existing methods to solve the task at hand…so as to match the needs of the user”. This is the case for Galileo and users of remote sensing in (1) the flexibility across possible inputs, (2) the accuracy across 10 diverse benchmarks, and (3) the effectiveness in the fine-tuning regime (for resource-rich groups) and kNN & linear regime (for resource-poor groups).
>
> ### 2. Technical Distinction between AllDisc and PatchDisc
> AllDisc samples negative examples from all patches in a _batch_, as opposed to within an _instance_. AllDisc can yield empirical gains compared to PatchDisc (e.g. 27% improvement on EuroSat, Table 6). We agree that AllDisc is a simple modification of PatchDisc. This simple modification that gives significant improvement is a strength of our method. We will update Section 2.2.1 to clarify this distinction.
>
> ### 3. The rationale behind the choice of PatchDisc and AllDisc
> We train Galileo with PatchDisc for local and global learning (as discussed in Section 2.2.3), but we observe significant improvements when using AllDisc for global learning (Table 6), so we include both in the text. We thank the reviewer for the feedback on the choice of PatchDisc vs. AllDisc in the final combined method, and we will make this clearer in the method section.
>
> ### 4. Masking Strategy and Depth Details
> Thank you for your attention to these important details. While space-time masking is best when learning global features (Table 6), random masking is best when learning local features (Table 7). Note that the target encoder depth also matters: prior work always used the full encoder to construct targets, but we find that a target depth of 0 is best for learning local features, and depth that varies per modality is optimal for learning global features. We are the first to vary target depth in this way, and the first to show its importance for remote sensing. We will clarify the use of random masking in Section 2.2.2 per this comment.
>
> ### 5. Performance and Complexity vs. CROMA
> Galileo-Base outperforms CROMA-Base on image tasks (Table 1) and is less complex (CROMA-Base has 60% more parameters than Galileo-Base). Galileo is architecturally simpler than CROMA, which requires 3 encoders to process a Sentinel-1&2 image pair (an MS optical encoder, a SAR encoder, and a fusion encoder). Galileo leverages a single encoder to process inputs across space, time, spectral bands, modalities, etc. Re: dataset size, CROMA was pretrained on SSL4EO with 1M samples of 264x264 pixel images while Galileo was pretrained on 127K samples of 96x96 pixels at 24 timesteps. In total, CROMA was pretrained on >2x as many Sentinel-2 pixels, so Galileo’s performance is not explained by dataset size.

---

> > ### Comment · Reviewer_1X8T · 2025-04-07
> >
> > The author's response has addressed most of my concerns, but the method novelty concern is still here. Therefore, I will upgrade my rating to weak accept.

---

### Official Review · Reviewer_yXVW · 2025-03-10

**Overall Recommendation:** 3

**Summary:**

The paper proposes a multimodal geospatial foundation model called Galileo. The authors also propose a new joint dataset combining various modalities with temporal, spatial, and spatiotemporal variations. As the architecture is ViT based the authors also provide methods for generating patches for diverse resolutions and also for adding spatiotemporal embedding. Representation learning is done on the latent space and the pixel space. The losses are based on the patch discrimination as in LatentMIM.
Evaluation is done on GeoBench and the method is compared to various other geospatial foundation models.

## update after rebuttal
Thank you for the clarifications. I will keep my score and still opt to accept the paper.

**Claims And Evidence:**

The general claims are supported by the evaluation.

**Essential References Not Discussed:**

I am not aware of any missing paper in the area.

**Experimental Designs Or Analyses:**

The experimental design seems appropriate

**Methods And Evaluation Criteria:**

The paper is evaluated on GeoBench which is the common benchmark for this type of models.

**Other Comments Or Suggestions:**

The paper often relates what is done to the strategies of other papers, which sometimes makes it harder to grasp what you have actually done. As you mostly justify your choices with the experimental results presented later on anyway, it sometimes would be more helpful to describe what you have in more detail.

**Other Strengths And Weaknesses:**

Strong points:
* The proposed method achieves convincing results on Geobench.
* The combination of modalities using varying embedding to generate uniform input token size for the ViT is elegant.
* The paper examines if mapping latent space representation for SSL makes sense for remote sensing tasks.

Weak points:
* It is unclear whether adding the variety of modalities is useful or hindering as the models were only evaluated on Sentinal2 tasks.
* The technical contribution is kind of using two known methods instead of just one which are alternated in training. Besides this, losses, masking strategies, and training methods are well-known from previous works in representation learning

**Questions For Authors:**

You propose AllDisc which is equivalent to PatchDisc except for averaging over the batch. I am wondering how PatchDisc computes a loss over a complete batch if not by averaging the loss. What is the exact difference for computing the loss over batch B?
(on 2nd thought, I guess the difference is the averaging in the denominator which is either over the batch or the patches within on image)
If you average over the batch, do you average over all the patches in the image as well?

The combination of patches from a large selection of modalities, times, and spatial patches seems to result in rather large amounts of maximal input tokens. How many tokens were used as maximal input and what were the GPU memory requirements resulting from it?
What was your masking percentage in training? Was it the same for both branches of your encoder?

**Relation To Broader Scientific Literature:**

Geospatial foundation models are currently trending and as far as I could see the authors relate their work to the SOTA in the area and uses the acknowledged Benchmark for this direction.

**Theoretical Claims:**

The paper does not contain any theoretical claims that need to be proven.

---

> ### Author Rebuttal · Authors · 2025-03-31
>
> Thank you for your review, and your detailed questions.
>
> ## Weaknesses
>
> ### 1. Evaluation on only Sentinel 2 tasks
> We agree that benchmark datasets over-represent Sentinel-2 based tasks, providing few opportunities to test the value of Galileo’s many modalities that reflect the wide diversity of input sensors used in the real world [1,2,3]. We benchmark on Sen1Floods11 (includes Sentinel-1 inputs), and CropHarvest (includes highly multimodal inputs, including topography and weather). Galileo excels on both of these datasets.
>
> We ablate all of our input modalities in Table 9 and find that - even for Sentinel-2 tasks - diverse modalities significantly helps performance (e.g. MADOS gains 4% with a model trained on VIIRS night lights compared to without).
>
> The above points show that the variety of modalities is useful, not a hindrance.
>
> ### 2. Technical novelty
> Our work introduces several technical innovations for self-supervised learning in general and pretraining models for remote sensing specifically:
>
> 1. We are the first to successfully fuse a diverse set of multiple sensors and products (e.g. weather) across space and time in a pretrained model. This fusion broadens Galileo's real-world applicability — as these inputs make a difference to downstream tasks [2,3,4] — and offers important empirical findings for multimodal self-supervision (e.g. the contributions of modalities to performance in Table 9).
> 2. We introduce contrastive learning with _varied depth_ targets, which we show is highly effective (8% improvement on the MADOS benchmark, Table 7). Our algorithms construct target tokens from (i) linear projections of the inputs (in our local loss) and (ii) a varying number of target encoder layers (in our global loss). Galileo is the first to exploit early-exited targets in SSL - we show its effectiveness via extensive ablations.
> 3. Galileo constitutes a novel combination of our dual local and global SSL losses which are each novel on their own. We demonstrate the efficacy of the algorithms in isolation (Tables 6, 7) and when combined (Table 8).
> 4. A novel combination of existing methods can also constitute technical novelty. For example, CROMA [NeurIPS ‘23] combined MAE and radar-optical contrastive learning: these SSL methods, _independently_, were not novel to remote sensing or ML in general. However, its _joint_ reconstructive and contrastive multi-modal self-supervisory method was novel.
> 5. We note that this work is submitted as “Application-Driven Machine Learning” as implemented by the [ICML 2025 process](https://icml.cc/Conferences/2025/ReviewerInstructions). This has different reviewing criteria including “Originality need not mean wholly novel method: It may mean a novel combination of existing methods to solve the task at hand…so as to match the needs of the user”. This is the case for Galileo and users of remote sensing in (1) the flexibility across possible inputs, (2) the accuracy across 10 diverse benchmarks, and (3) the effectiveness in the fine-tuning regime (for resource-rich groups) and kNN & linear regime (for resource-poor groups).
>
> ## Questions
>
> ### 1. AllDisc vs. PatchDisc
> AllDisc samples negative examples from all patches in a _batch_, as opposed to within an _instance_. AllDisc yields significant empirical gains compared to PatchDisc (e.g. a 27% improvement on EuroSat, Table 6). We will make the following change to the “Loss function” part of Sec. 2.2.1 to clarify this (updates in italics): "To encourage globally discriminative representations, we extend the PatchDisc loss to better discriminate samples in a batch. _We achieve this by sampling negative examples from the entire batch, as opposed to within the sample._"
>
> ### 2. Maximal input tokens
> We trained all the final Galileo models on a single GPU with 80Gb of memory, but smaller mini-batches allowed us to run many of our pretraining experiments on a consumer-grade 24Gb GPU. We agree that a large number of tokens is a challenge when incorporating many modalities, timesteps and spatial dimension: when subsampling the inputs (Appendix B.2) we selected (size, timestep) combinations so that the maximum number of input tokens was 1500. The masking percentage during training was 10% of tokens unmasked and 50% of tokens decoded; this was consistent for both the global and local branches.
>
> ## Other comments
>
> ### 1. Relating strategies to past papers
> Thank you for this feedback. To better balance our own descriptions with relations to other work, we will first provide a self-contained summary of our method at the start of Sec. 2 (and we will move the preamble on motivations to the supplementary material to make space for this suggestion).
>
> [1] https://arxiv.org/abs/2312.03207
>
> [2] https://essd.copernicus.org/articles/15/5491/2023/
>
> [3] https://soil.copernicus.org/articles/7/217/2021/

---

> > ### Comment · Reviewer_yXVW · 2025-04-08
> >
> > Thank you for the clarifications, I will keep my score .

---

### Official Review · Reviewer_ZDMq · 2025-03-11

**Overall Recommendation:** 3

**Summary:**

This paper proposes the "Galileo" family of pre-trained remote sensing models, which aims to learn both global and local features to cope with the multimodal, variable input size, and large-scale span characteristics of remote sensing data. The authors improve the ViT architecture to enable the model to flexibly handle inputs from multiple sensors, and design a self-supervised learning algorithm that uses global and local objectives to learn coarse-grained and fine-grained features, respectively. The paper provides a large number of benchmark experiments, ablation studies, and comparative experiments.

## update after rebuttal
Thank you for the clarifications. I will keep my score.

**Claims And Evidence:**

The authors claim that Galileo model can flexibly handle different modalities, inputs of different sizes, and targets of different scales.
To demonstrate this, the authors provide detailed comparative experiments and ablation experiments.

**Essential References Not Discussed:**

This part makes sense.

**Experimental Designs Or Analyses:**

The experimental part is sufficient and complete.

**Methods And Evaluation Criteria:**

This part makes sense.

**Other Comments Or Suggestions:**

The author proposed an SSL algorithm that focuses on low-frequency and high-frequency features. The introduction, methods and appendix do not reflect related research in the image frequency domain.

**Other Strengths And Weaknesses:**

Advantages:
1. The issues and challenges that the paper focuses on are meaningful.
2. There is a wealth of experimental support for this model.
Weakness:
1. The global and local pre-training objectives are simple and clear. The algorithm does not seem to be able to balance the two well, and the optimization for multiple objects is not well explained.
2. The operations on input scale and shape seem to be resizing techniques and image preprocessing techniques.

**Questions For Authors:**

1. The author subsampled the modalities and input shapes from the dataset to construct realistic scenes. Does this construction match the real world? After that, patchification and sampling related project methods are used for different shapes and scales. The whole process seems to be downsampling the dataset and then upsampling. Could the authors explain the scale issue semantically?

2. The author focuses on processing timesteps-related data. However, the paper is about classification and semantic segmentation. Is "time" important in this field? Are there any experiments on tasks related to time series such as change detection?

**Relation To Broader Scientific Literature:**

This paper is motivated by the input shape and scale issues of remote sensing images, which inspires the study of foundation models in remote sensing.

**Theoretical Claims:**

This paper proposes a pre-trained multimodal framework. The main improvements are based on the data and network structure. There is no problem with the theoretical algorithm.

---

> ### Author Rebuttal · Authors · 2025-03-31
>
> Thank you for your thorough review; we are glad you recognize the meaningful challenges we address with Galileo, since this was one of the primary objectives of this work.
>
> ### Balancing the losses
> We combine the global and local objectives via a simple average (Section 2.2.3), and demonstrate via ablations that Galileo can learn from this combination effectively (Table 8). There is no balancing of the losses, and no multi-objective tuning to explain.
>
> We also measure token representation similarities in Table 2: combined retains the within-sample diversity of local and achieves between-sample diversity between global and local. The optimal diversity within or between samples is unknown and likely task-dependent; we offer these measurements to complement our benchmark evaluations.
>
> ### Operations on input scale and shape / resizing + image processing
> Galileo is the first to harmonize multi-modal and multi-resolution inputs for remote sensing in this way. Relative to FlexiViT, we modified the model architecture (“Flexible input shapes”, Section 2.1.2) and the training recipe (described in detail in Appendix B.2). Remote sensing analyses may require data as pixel timeseries [1] or imagery [2] across many sizes and shapes; Galileo is the first model that can process any of these diverse inputs for a range of spectral bands and other products (e.g. weather, topography).
>
> ### Lack of related research in the image frequency domain
> We cite remote sensing papers that aim to learn both high and low frequency features, including ScaleMAE and SatMAE++ (which include in our evaluations in Tables 3 and 4). We welcome the suggestion of additional references.
>
> ## Answers to questions
>
> ### 1.a. Does this construction match the real world?
> Yes: different real world applications of machine learning for remote sensing use very different input shapes, which reflects our subsampling. For example, Skylight [2] uses single-timestep S2 imagery or multi timestep S1 imagery, WorldCereal [1] uses highly multimodal (S2, S1, topography, weather) pixel timeseries, and Global Plastics Watch [3] uses both images and pixel-timeseries. **Galileo is the first model that can support these diverse, real world use cases**.
>
> ### 1.b. Could the authors explain the scale issue semantically?
> Galileo is the first pretrained model to incorporate inputs of significantly different resolutions (e.g. ERA5 at ~30km/pixel vs. Sentinel-2 at 10m/pixel). For inputs significantly coarser than 10m/pixel, we treated them as unchanging in space (i.e. as a pixel timeseries).
>
> We use the term “multiscale” to describe the scale of the targets (e.g. vessels, which occupy a few pixels, vs. glaciers which span kilometres). We apologize for the confusion and will clarify this in the preamble to Section 2.
>
> ### 2.a. Is "time" important in this field?
> Yes: previous work has found the temporal dimension to be critical, e.g. for agricultural land cover mapping like our PASTIS benchmark (and even more important than the spatial dimension) [4]. Many large scale mapping efforts (i.e. segmentation) model pixel timeseries to focus on the time dimension instead of the spatial dimension [2, 5, 6].
>
> ### 2.b. Are there any experiments on tasks related to time series?
> Our paper contains 3 evaluation datasets with a temporal dimension, two of which only contain the temporal dimension. Results for PASTIS (agricultural land cover segmentation) are in Table 4. Table 1 of [7] found that ignoring PASTIS’s time dimension performed significantly worse. Results for the CropHarvest and Breizhcrops pixel timeseries tasks are in Table 5. Galileo is the best or second-best method across time-series datasets.
>
> [1] https://essd.copernicus.org/articles/15/5491/2023/
>
> [2] https://arxiv.org/abs/2312.03207
>
> [3] https://journals.plos.org/plosone/article?id=10.1371/journal.pone.0278997
>
> [4] https://arxiv.org/abs/1901.10503
>
> [5]  https://soil.copernicus.org/articles/7/217/2021/
>
> [6] https://esa-worldcover.org/en
>
> [7] https://arxiv.org/abs/2107.07933

---

> > ### Comment · Reviewer_ZDMq · 2025-04-05
> >
> > Thanks for the reply. I will keep this score.

---

### Official Review · Reviewer_VuYX · 2025-03-14

**Overall Recommendation:** 3

**Summary:**

This paper introduces *Galileo*, a family of pretrained ViTs that flexibly encode multi-source Earth observation (EO) data of varying spatial and temporal scales for various downstream tasks. To address limitations in existing pretrained EO "foundation models", *Galileo* uses a self-supervised learning (SSL) recipe inspired by I-JEPA to simultaneously learn large-scale global features suitable for coarse-grained tasks like image classification and small-scale local features ideal for dense prediction tasks. The proposed approach includes a latent prediction task formulated in a contrastive manner to achieve discriminative intra-image and inter-image patch representations. Through additional techniques such as *FlexiViT*, dynamic encoder depth, and structured masking, *Galileo* achieves flexibility in input resolution, representation granularity, and available input sources. Extensive experiments demonstrate *Galileo*'s performance compared to various baselines across multiple EO tasks, including image classification, timeseries classification, semantic segmentation, and timeseries segmentation.

## Update after Rebuttal
After the author's response to my questions, my main concerns about the construction of the proposed method and the evaluation protocols have been resolved. In addition, given the comprehensiveness of the experiments, I raised my score to 3.

**Claims And Evidence:**

1. **Flexible input shapes**: Although FlexiVit has been validated to perform well for various patch sizes through ImageNet benchmarks, the author’s claim on input shape and patch size flexibility can be greatly improved by adding evaluations to demonstrate the model’s performance change with respect to multiple input sizes. An example of this is the Figure 4 of FlexiViT.
2. **Benchmark Task**: Although EuroSat has been widely used in various prior works about “remote sensing foundation models,” I find this benchmark unconvincing, as a simple ResNet can easily achieve 99% accuracy [1, 2]. Evaluating Galileo against other models on a challenging and realistic benchmark, such as Fields of the World (FTW) [2], in Figure 3, can provide more convincing evidence for Galileo’s usefulness.
3. **Baseline methods**: I recognize the authors for the extensive comparisons with prior works. However, to demonstrate the usefulness of the work, it would be helpful to compare Galileo with a few specialized models (such as ResNet and SwinViT for monotemporal tasks and TSViT [3] for timeseries tasks) trained from scratch on the evaluation task similar to GeoBench [2]. In addition, a comparison to SatCLIP [4] on certain image-level tasks would also be beneficial.
4. **Contrastive learning in the pixel space**: As Galileo still performs patchification in the local I-JEPA pretraining task, I find the claim of being “the first SSL algorithm to perform contrastive learning in the pixel space” unsubstantiated, although a patch size of one can be randomly selected as suggested in Appendix B.2. In addition, the downstream evaluation uses “a patch size of 4 for all models with variable patch sizes”, which also does not help justify the usefulness of “contrastive learning in the pixel space.”
5. **Effectiveness of the proposed loss**: In Table 8, PatchDisc only has marginal improvements over MSE, so I think this is a rather weak signal for practitioners to choose an I-JEPA-like paradigm against paradigms like BYOL and SimSiam in which features are matched with MSE without the overhead of introducing negative examples.

[1] https://paperswithcode.com/sota/image-classification-on-eurosat

[2] Lacoste, A., Lehmann, N., Rodriguez, P., Sherwin, E., Kerner, H., Lütjens, B., Irvin, J., Dao, D., Alemohammad, H., Drouin, A. and Gunturkun, M., 2023. Geo-bench: Toward foundation models for earth monitoring. Advances in Neural Information Processing Systems, 36, pp.51080-51093.

[2] Kerner, H., Chaudhari, S., Ghosh, A., Robinson, C., Ahmad, A., Choi, E., Jacobs, N., Holmes, C., Mohr, M., Dodhia, R. and Ferres, J.M.L., 2024. Fields of the world: A machine learning benchmark dataset for global agricultural field boundary segmentation. arXiv preprint arXiv:2409.16252.

[3] Tarasiou, M., Chavez, E. and Zafeiriou, S., 2023. Vits for sits: Vision transformers for satellite image time series. In Proceedings of the IEEE/CVF Conference on Computer Vision and Pattern Recognition (pp. 10418-10428).

[4] Klemmer, K., Rolf, E., Robinson, C., Mackey, L. and Rußwurm, M., 2023. Satclip: Global, general-purpose location embeddings with satellite imagery. arXiv preprint arXiv:2311.17179.

**Essential References Not Discussed:**

N/A

**Experimental Designs Or Analyses:**

Yes, kNN and linear probing with varying training dataset size is a common approach to evaluate pretraining image encoders. I left other comments about the selection of downstream tasks in *Claims and Evidence*.

**Methods And Evaluation Criteria:**

The proposed method extends prior works in computer vision such as FlexiViT and I-JEPA. They make sense on a high level. The evaluation criteria follow widely recognized benchmarks in the domain.

**Other Comments Or Suggestions:**

1. I find it hard to parse Figure 1 and Figure 2 together. For example, it took me a while to figure out x1 is for the “global” task with early exit in Figure 2 and  x3 is the label for the “local” task.
2. Terms such as "local/global," "shallow/deep," and "high-level/low-level" are used almost interchangeably. Given that the local and global features/tasks in this paper (feature abstraction levels determined by the depth of the encoding modules) are different from those in other literatures (spatial locations), consider choosing and consistently sticking to one pair of terms to avoid confusion and provide the reader with more explanation about the terms.

**Other Strengths And Weaknesses:**

The strength of the paper lies in the extensive experiment design. The clarity of the writing, notations, and figures should also be greatly improved before being accepted for publication. I outline other weaknesses in *Claims and Evidence*, *Other Comments Or Suggestion*, and *Questions For Authors*.

**Questions For Authors:**

1. The authors claim that “ours is the first SSL algorithm to perform contrastive learning in the pixel space.” Does this mean that the patch size for the local pretraining task is always one?
2. Can we use a patch size of one for segmentation tasks? Will it yield substantial improvements over a patch size of four?
3. In Table 12, Galileo does not outperform DeCUR on m-BigEarthNet and does not outperform CROMA on m-Brick-Kiln. Does the author have any explanation or intuition for this observation?
4. In the definition of AllDisc, is the softmax temperature learned, following a schedule, or fixed? I could not find other references to the temperature in the paper except for this definition.
5. In the local and global I-JEPA tasks, both the predictor and the encoder receive the same (position, time, and channel group) embeddings (in the predictor cross attention and after patch embedding layers). Does this shared information create potentially shortcuts in the contrastive loss?
6. Could the authors further justify adding the raw location of the image in training? Could it increase the risk of overfitting? Is using raw coordinates instead of functional positional encoding optimal for encoding locations? [1]
7. With the modifications to I-JEPA, do the authors have any observations about the effect of batch size on performance since the negative examples now include patches from other training examples?
8. Could the authors clarify which input sources are considered targets? Is the online encoder receiving the same inputs but with different patchification and masking strategies for the global and local tasks?

[1] Klemmer, K., Rolf, E., Robinson, C., Mackey, L. and Rußwurm, M., 2023. Satclip: Global, general-purpose location embeddings with satellite imagery. arXiv preprint arXiv:2311.17179.

**Relation To Broader Scientific Literature:**

Existing pretrained remote sensing models (e.g., SatMAE (Cong et al., 2022), CROMA (Fuller et al., 2024), MMEarth (Nedungadi et al., 2024), AnySat (Astruc et al., 2024)) focus on single or a specific combination of modalities or limited input shapes. Galileo explicitly designs the transformer architecture to handle varied input modalities (multispectral optical, SAR, topographic, temporal data) and input dimensions by using a customized tokenization approach that can handle arbitrary combinations of modality, temporal steps, and spatial resolutions.

**Theoretical Claims:**

N/A

---

> ### Author Rebuttal · Authors · 2025-03-31
>
> Thank you for the thorough review and excellent suggestions.
>
> ## Claims and Evidence
> ### 1. Flexible input shapes
> Fig 3 and Tab 11 show Galileo performs well across varying patch sizes. Tabs 3 and 5 show Galileo takes both pixel timeseries and images. We also run MADOS segmentation with a smaller patch size of 2 [here](https://imgur.com/a/JMEjrvb).
>
> ### 2. Benchmark task
> We agree that EuroSat alone is insufficient. We showcase SoTA results on **ten diverse datasets** incl. FloodBase Sen1Floods11 and NASA CropHarvest, created by real-world practitioners.
>
> FTW [AAAI 25] is concurrent per ICML policy. FTW cite PASTIS as a “relevant dataset for comparison"; we evaluate on PASTIS in Tab 4.
> ### 3. Baselines
> We show that Galileo is the best on average against 16 other pretrained RS models, which in turn gain over specialized models, e.g. SatMAE outperforms a fully supervised ResNet50 (their Tab 9) and Presto outperforms a SITS Former (their Tab 6).
>
> We run new finetuning results [here](https://imgur.com/a/NkJlUOD) with a SwinViT (Satlas) as requested. We update our rankings (Tab 14) [here](https://imgur.com/a/Dkz8Lgk) w/ finetuning: Galileo-Base has the best average rank.
>
> SatCLIP [AAAI 25 = concurrent work] is a location encoder (a function of lat/lon coordinates) while Galileo is a visual encoder (a function of image/pixel spatial/temporal inputs). SatCLIP only compares itself to other location encoders.
> ### 4. Contrastive learning in the pixel space
> This describes our use of targets from linear projections of input pixels (this work) instead of from deep representations (prior work). See Sec 2.2.2 at “Target Depth” and Fig. 2 (step 4). We thank you for highlighting the insufficient clarity of this novelty: we will edit Sec. 2.2.2 to explain. Galileo is the first model to exploit contrastive learning on _multiple depths_ in SSL - we show its effectiveness via extensive ablations (e.g. 8% gain on MADOS: Tab 7).
> ### 5. Proposed loss
> PatchDisc losses outperform MSE losses by 5.4% on MADOS, 0.5% on Sen1Floods11, 1.6% on CropHarvest, and 2.3% on EuroSat (Tab 8). These controlled ablations verify the value of our loss for pretraining on remote sensing data.
>
> ### "Extension of FlexiViT and I-JEPA"
> Galileo is not simply an extension: it introduces 1) a first-of-its-kind encoder for our set of more general inputs, 2) a new latent masked modelling method with _multiple depth_ targets, 3) a novel combination of two methods, which are themselves novel (our local and global losses). We describe these in detail [here](https://openreview.net/forum?id=gqZO3eSZRy&noteId=SQm6v8ByKG).
>
> ## Questions
> 1. **Local task**: Please see "contrastive learning in the pixel space" above.
> 2. **Patch size**: Please see “flexible input shapes” above. We provide new results for patch size = 2 as an alternative to patch size = 1 ( size = 1 results in too many embeddings and excessive computation).
> 3. **Galileo vs. DeCUR, CROMA**: No one model is best for all tasks, but Galileo is best on average (please see the rankings [here](https://imgur.com/a/Dkz8Lgk)). CROMA and DeCUR’s image specialization may help on m-BigEarthNet and m-Brick-Kiln.
> 4. **AllDisc loss temp**: We fix the temp and will note this in Sec. 2.2.1.
> 5. **Contrastive shortcuts**: Great observation. Our local loss prevents this shortcut, as position/channel/time embeddings are not added to our input projections. Our combined algorithm stabilizes training (100% of runs achieve >80% on EuroSat in Tab 8 vs. 63% in Tab 6). We emphasize these are not simply I-JEPA losses ([link](https://openreview.net/forum?id=gqZO3eSZRy&noteId=SQm6v8ByKG)).
> 6. **Location inputs**: Galileo takes optional location inputs and achieves top or near-top benchmark results (Tab. 3-5) without them (only CropHarvest has location inputs); see our new result (last row) in [link](https://imgur.com/a/RJk1BAJ). We include locations because they are included in real world applications [1].
> 7. **Batch size effects**: AllDisc experiments used the largest batch size that fits in memory (a common heuristic). As requested, we run our global feature learning algorithm with a smaller batch size ([link](https://imgur.com/a/RvkPB9w)). A smaller batch size doesn't hurt performance.
> 8. **Inputs and targets**: Correct, all data sources are inputs and targets for both the global and local tasks. Patch sizes are randomly sampled from the same distribution for the global and local tasks. We will edit Sec. 2.2.1 and 2.2.2 to reinforce these points.
>
> ## Other comments
> 1. **Hard to parse Figs 1, 2**: Thank you for highlighting this. We will edit Fig 1 to relate the (x_1, x_3) paths to the local and global tasks in Fig 2 and simplify Fig 2 by removing the grid in step 5
> 2. **[terms] are used almost interchangeably**: Thank you for highlighting this potential confusion. We will standardize terms to use “local/global” for loss targets and branches vs. “shallow/deep” for target depths.
>
> [1] https://essd.copernicus.org/articles/15/5491/2023

---

> > ### Comment · Reviewer_VuYX · 2025-04-04
> >
> > I thank the authors for the follow-up experiments. As the response clarified my questions, I have changed my recommendation to acceptance.

---

### Decision · Program_Chairs · 2025-05-01

**Decision:**

Accept (poster)

**Comment:**

This paper introduces Galileo, a family of trained ViTs that flexibly encode multi-source Earth observation (EO) data of varying spatial and temporal scales for various downstream tasks. Extensive experimental evaluations have been conducted against baselines on multiple datasets. The main contribution is a pre-trained model for remote sensing datasets in various tasks. The authors' rebuttal has addressed reviewers' main concerns in model evaluation and comparison with related works. The authors are suggested to incorporate these revisions.